# Using simulation model as a tool for analyzing bus service reliability and implementing improvement strategies

**Seyed Mohammad Hossein Moosavi**[1]☯*, **Amiruddin Ismail**[2]☯, **Choon Wah Yuen**[1]☯

**1** Faculty of Engineering, Centre for Transportation Research (CTR), University of Malaya (UM), Kuala Lumpur, Malaysia, **2** Faculty of Engineering, Smart and Sustainable Township Research Centre, National university of Malaysia (UKM), Bandar Bangi, Malaysia

☯ These authors contributed equally to this work.
* mh.moosavi@um.edu.my, mh.moosavi65@gmail.com

**Data Availability Statement:** All relevant data are within the paper and its Supporting Information files.

**Funding:** The authors would like to acknowledge the Centre for Transportation Research and

## Abstract

Bus services naturally tend to be unstable and are not always capable of adhering to schedules without control strategies. Therefore, bus users and bus service providers face travel time variation and irregularity. After a comprehensive review of the literature, a significant gap was recognized in the field of public transportation reliability. According to literature, there is no consistency in reliability definition and indicators. Companies have their own definition of bus service reliability, and they mostly neglect the passengers' perspective of reliability. Therefore, four reliability indicators were selected in this study to fill the gap in the literature and cover both passengers' and operators' perceptions of reliability: waiting time and on-board crowding level from passengers' perspective, and headway regularity index at stops (HRIS) and bus bunching/big gap percentage from operators' perspective. The primary objective of this research is to improve the reliability of high frequency of bus service and simulation tools currently being used by the public transportation companies. Therefore, a simulation model of bus service was developed to study the strategies to alleviate it. Four different types of strategies were selected and implemented according to Route U32 (Kuala Lumpur) specifications. Model out-put showed that control strategies such as headway-based dispatching could significantly improve headway regularity by almost 62% and the waiting time by 51% on average. Both holding strategies at key stops (previous and Prefol holding) have shown an almost similar impact on reliability indicators. Waiting time was reduced by 44% and 43% after the previous and Prefol Headway strategies were adopted, respectively. However, the implementation of the component of headway-based strategies at the terminal and key stops showed the best impact on reliability, in terms of passenger waiting time. Waiting time and excess waiting time were both significantly reduced by 52.86% and 81.44%, respectively. Nevertheless, the strategies did not show any significant positive effect on the level of crowding during morning peak hours.

Institute of Research Management and Services University of Malaya for providing research facilities and funding under the research grant IIRG009A-19SAH. The funders had no role in study design, data collection and analysis, decision to publish, or preparation of the manuscript.

**Competing interests:** The authors have declared that no competing interests exist.

# Introduction

Urban public transport is an essential transportation mode because of the congestion on urban streets that results from the growing numbers of cars in cities. The demands for public transport in urban areas are growing, and technological efforts are being made to improve the services offered by urban transport networks. However, some studies indicate that technological methods are not capable of offering a reliable system since passengers' behavior can considerably affect service regularity [1,2].

It is very important to clearly understand the meaning of "bus service reliability" from different perspectives. Peek and van Hagen (2002) [3] developed a methodology to evaluate passengers' satisfaction and priorities, according to Maslow's pyramid. It is confirmed that security and reliability are fundamental definition of satisfaction for passengers, therefore must be provided [4–6]. The Transport Focus reported in March 2016 that bus industry in England is facing severe challenges. In fact, the number of passenger journeys in outside London continued to decrease [7]. Therefore, transport focus conducted a survey to evaluate passengers' perspective of bus service reliability. The report confirmed the significance of providing a frequent, punctual and reliable service that provides value for money.

It is demonstrated that study of service reliability is one of the most crucial part of urban transport improvement. Therefore, valuable studies were conducted on reliability definition, indicators and improvement strategies. For instance, Van Oort (2014) [8] conducted valuable surveys to evaluate service reliability from bus companies' perception. The results showed that companies pay little attention to service reliability in the initial stage of planning. Although the surveys provide useful information about how bus companies deal with service reliability, passengers' perception of bus service reliability was not considered in the surveys. In addition, Diab, Badami and El-Geneidy (2015) [9] conducted a review on bus service reliability from both passengers and bus companies' perspective. Moreover, strategies that bus companies applied in order to improve service reliability are included on their study. However, they didn't review main causes of service unreliability and factors negatively affect service regularity.

This section presents an introduction and review of literature on 1) the definitions of bus service reliability, 2) strategies to improve service reliability, and 3) simulation models as a tool to analyze service reliability. In the first step, we need to understand the definitions and indicators of bus service reliability clearly. Bus agencies also need reliable instruction to measure the level of bus service reliability. Therefore, we tried to answer these two questions in the first part of the introduction: 1) What is the most comprehensive definition of bus service reliability, and 2) How we can measure the level of reliability, considering both passengers' and companies' perspectives?

After understanding the definition of reliability, we need to study the methods and strategies for improving the level of reliability. Therefore, the second part of the literature was conducted pertaining to the most effective strategies to improve reliability. In order to evaluate the effect of strategies on the reliability of bus services, we need a simulation model. The third part of the literature review is focused on the existing simulation models. Section 2 presents the data collection procedures and a conceptual framework of this study. Moreover, the simulation model development and bus service reliability indicators will be discussed in this section. Results of implementation of the strategies and output of the simulation model will be presented in Section 3.

## Bus service reliability: Definitions and indicators

One of the most significant features of transit service quality is reliability. It is a major concern for transit agencies and passengers [10]. [11] defined reliability as regular headway, schedule

adherence and stability of travel times. [12,13] associated reliability primarily with keeping schedules and minimizing schedule-related delays, i.e., maximizing on-time performance (OTP) and minimizing run-time delays and variations and headway delays and variations. This was in line with conclusions drawn by [14,15].

The Transit Capacity and Quality of Service Manual (TCQSM, 2010) provided an extensive list of transit reliability measure instances. The manual gives particular attention to headway adherence and on-time performance, which are the most common reliability measures used in the transit industry [16]. The 2000 edition of the Highway Capacity Manual (HCM) introduced various level-of-service (LOS) measures for auto, transit, bicycle, pedestrian modes of transport [17]. Chapter 27 in HCM (2000) outlines the quality evaluation of the following four main transportation modes in terms of: a) service frequency, b) hours of service, c) passenger load, and d) service reliability. Chapter 17 of the same manual suggests that excessive waiting times reflect diminished transit vehicle reliability.

The TCQSM (2010) introduced a novel approach to measuring the LOS of the transportation system that covered two important aspects of LOS (comfort and availability) at three levels (stops, route segments, and the system as a whole). [18] analyzed this framework using AVL data that were collected in Trieste, Italy. According to the authors, the main limitations of the TCQSM method were that it considered the number of delayed trips but not the amount of delay, that it did not assess the impact of early departures on passengers, and that it had a fixed tolerance of the schedule in assessing the on-time performance (OTP).

Fu and Xin (2007) [19] proposed a transit service indicator (TSI) that evaluates the level of service by considering the impact of supply on demand. The TSI utilizes different quality metrics at the same time, such as frequency and coverage. Van Oort et al. (2012) [20] proposed how to increase reliability by adjusting schedule timetables through the usage of holding points. The authors used punctuality, defined as on-time arrival, together with the probability of on-time departure from the terminal, to measure reliability. Chen et al. (2009) [21] developed three parameters related to performance to evaluate the reliability of bus services: a punctuality index based on routes (PIR), a deviation index based on stops (DIS), and an evenness index based on stops (EIS). The authors showed that these parameters revealed low bus service reliability in Beijing and presented correlations between service reliability and route length, headway, and distance from the stop to the origin terminal, as well as the effect of providing separate bus lanes.

According to currently available literature, passenger's definition of reliability is more focused on minimizing travel time, which includes off-board and on-board waiting times. However, some researchers had concluded that the value passengers put on running time regularity is higher than the value that they put on travel time [22–24]. The passenger load is another indicator that can significantly affect the users' experience and service quality. It can decrease the satisfaction level of users by negatively affecting their mental and physical health [25–30].

On the other hand, agencies mainly considered schedule adherence or on-time performance (OTP) as a reliability indicator. Constant headways and running time regularity were additional factors of perennial concern to agencies. It should be mentioned that it was challenging for agencies to evaluate and quantify the passengers' level of satisfaction. Most service providers used surveys to measure passengers' perspectives [9]. These surveys were developed by experts to evaluate passengers' satisfaction, identify causes of unreliability, and gain insights into policies and strategies for improving the service quality. The key findings concerning reliability indicators from two different perspectives (passengers and service providers) are illustrated in Fig 1. The main difference between these two points of view is related to the definition of reliability.

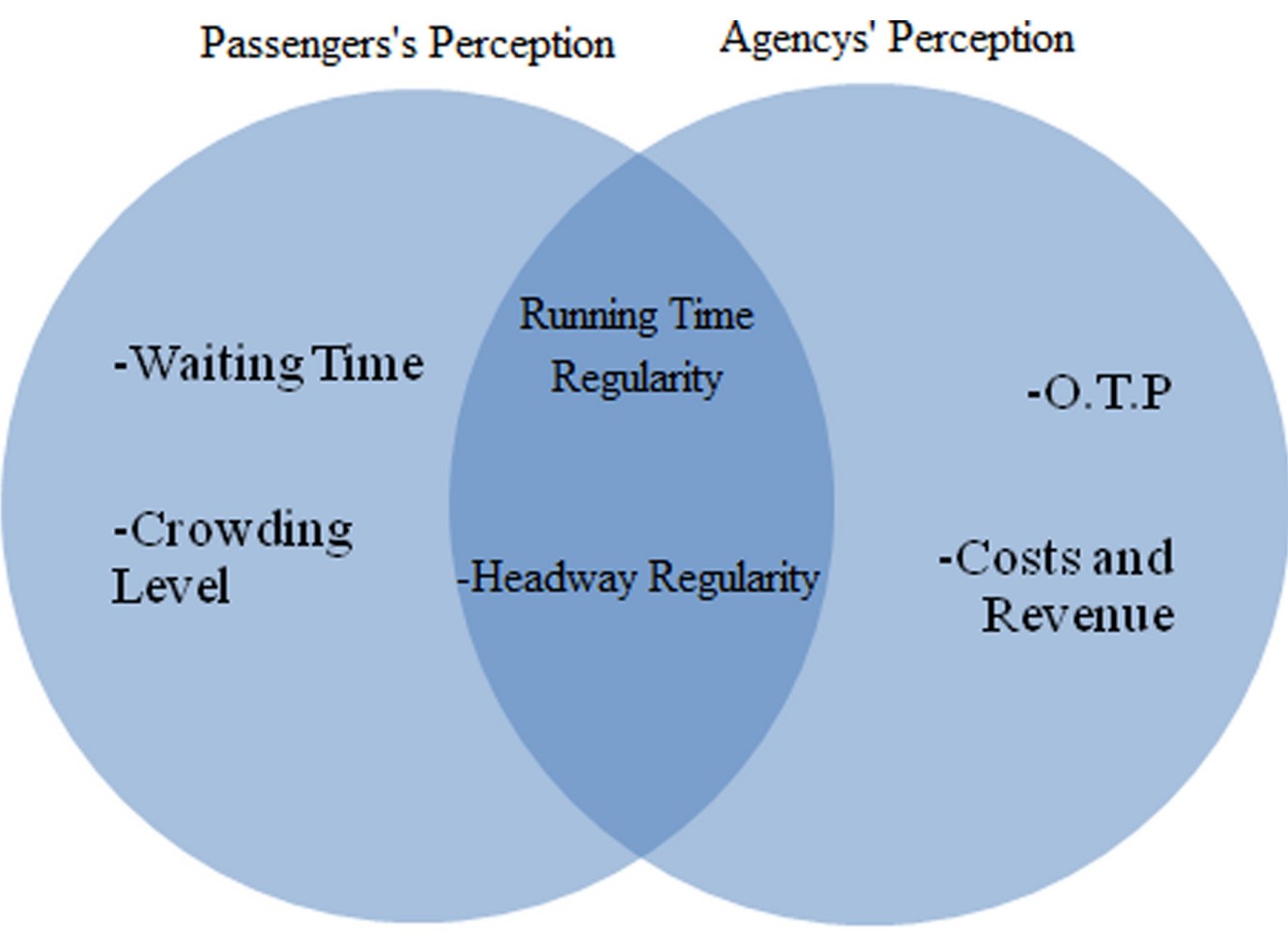

**Fig 1. Reliability definition from passengers' and agencies' point of view.**

### Strategies to improve bus service reliability

As mentioned before, bus services, in general, tend to be unstable. Bus services cannot adhere to a schedule without control strategies and will exhibit bunching or big gaps eventually. There are some conventional strategies that operators use to hold buses at specific control points. These strategies reduce the likelihood of bus bunching but require considerable slack to slow down the vehicles. These methods increase the in-vehicle passenger waiting times and operation costs [31]. Operation control methods seek to optimize system performance in cases of interruption of service, depending on the current system status [32]. Turnquist and Blume (1980) [11] argued that there is a difference between the planning of service and strategies to control service in real-time. Strategies related to the planning of service focus on persistent problems and achieve solutions by changing schedules and restructuring routes. In contrast, control strategies employed in real-time need to provide instantaneous solutions to problems that occur suddenly [33].

According to [34], priority, control, and operational methods are used to enhance the reliability of the bus service. Priority methods, such as providing separate lanes for buses and prioritizing traffic signals for bus movements, involve paying special attention to buses instead of the general traffic. Operational methods involve long-term strategies, including education of drivers, changes in schedule, and restructuring of bus routes. Control methods are short-term

actions taken in real-time, including modification of speeds, suggestions to skip certain stops, and short-turning and holding of certain buses. With respect to operation control, there is a crucial distinction between preventive and corrective strategies. The difference stems from the time of adoption of the strategy and its goal. Corrective strategies are typically employed to prevent bunching when an interruption has occurred. Preventive strategies are applied before an incident occurs. One such strategy would be to plan in advance to prevent significant deviations in headway. To develop an efficient preventive strategy, a planner has to have effective estimation skills and knowledge of the operating conditions [35].

In developing operation control strategies, high-frequency and low-frequency routes must be considered independently. Logically, for low-frequency routes, strict adherence to the schedule is of the utmost importance, as it is assumed that passengers arrive at the bus stops based on a schedule [36,37]. However, if the headway is less than 10 min (high-frequency), maintaining headway takes priority over strict adherence to the schedule, because it is assumed that passengers arrive randomly at the stops of high-frequency routes.

An effective preventive strategy is provided by evaluating network designs at the initial stages [38]. By using real-time data, it is possible to control headway deviation and enable the adoption of preventive actions aimed at avoiding probable service irregularities. Holding is one the most common corrective used strategies between service providers. Holding strategies consist of holding a bus in one selected location (key stops) on propose, to decrease the waiting time of passengers on the next stops. However, holding will increase the travel times for onboard passengers. Therefore, terminal is one of the most desirable locations for implementing holding strategy [31,39]. According to previous studies, holding strategy is the most effective strategy, and several methods have been proposed (from simple heuristics to sophisticated model-based optimization) in order to evaluate the holding intervention. Previous research has established that holding is generally the single most effective type of intervention. A number of methods to determine the holding policies have been proposed and evaluated, ranging from simple heuristics to sophisticated model-based optimization [40,41]. Number of studies present various headway-based strategies for stabilizing the irregularity in bus services [10,31,42–44]. In addition, Berrebi et al. (2015) [45] suggested a real-time dispatching strategy in order to minimize the waiting time. However, these studies had a weakness: they considered buses as a separated system and did not take the impact of traffic and signal control into account. Chow et al. (2017) presented a set of optimal control strategies in order to improve bus service reliability by implementing adjustments on signal timings [46].

When two or more buses arrive at a bus stop at the same time or in a very short time interval is called bus bunching. Various methods have been suggested to overcome this problem in bus routes in order to improve service regularity [47–52]. Fonzone et al. (2015) stated that one of the main reasons for bus bunching could be insufficient boarding rate when overall passenger demand is considered [53]. Yu et al. (2016) developed a method to detect bus bunching using smart card data archive [54]. A new strategy was developed by Cao and Ceder (2019) to optimize the bus service timetable by using the stop-skip method, based on real-time passenger demands [55]. In addition, some studies proposed speed adjustment policy to alleviate bus bunching in dedicated bus lanes [56–58].

In this study, two different types of strategies are implemented at the terminal: Scheduled-based departure and headway-based departure (as shown in Table 1). In scheduled-based departure, buses are forced to depart from the terminal strictly on-time. This strategy would be possible through effective supervision at the terminal. Four different types of strategies and their component were implemented on Route U32, and the results were recorded by a simulation model.

**Table 1. Description of strategies.**

| Strategies | Description | Location |
|---|---|---|
| **Strategy 1** | **Schedule-based departure from terminal** | **Terminal** |
| Strategy 2 | Headway-based departure from terminal | Terminal |
| Strategy 3 | Previous holding strategy | Key stops |
| Strategy 4 | Prefol holding strategy | Key stops |

## Simulation model as a tool to analyzing bus service reliability

Public transportation is a very extensive and complex system. Thus, researchers will not be able to develop theoretical methods for such a system without adapting simulation models. Moreover, it is impossible to directly implement any experimental strategies on a route in the real world, due to the high risk of waste of time and capital for both companies and passengers. Therefore, simulation models and analysis tools are needed in order to evaluate public transport systems and improvement strategies [59].

The behavior of passengers, the dynamics of road traffic, and particular bus network operations are the key and interactive elements of any bus network system. The interactions of the elements mentioned above make the bus network system complex. Some studies have shown that the multi-agent approach is the most appropriate for explaining a bus network system and the interactions between its key elements [60–63]. The multi-agent approach is based on two assumptions. The first is that the public transport network in any city can be defined as a complex system consisting of interacting entities [64–66]. The second is that the behavior of a global system consists of phenomena derived from the behavior of separate entities, as well as from the interactions among the entities [67–69].

Andersson et al. (1979) [70] proposed an interactive simulation model to assess a city bus route during a peak traffic jam. This model can serve as a tool and guide for route control operators by helping them to understand how each individual act impacts the service. The simulation model developed by Abkowitz and Tozzi (1987) [71] can be used to evaluate how effective timed transfer is in a transit operation based on a schedule. This is accomplished through a case study of the imagined routes that cross at a transit key stop. Chandrasekar et al. (2002) [72] developed a microsimulation model that can be used to assess the efficiency of linked strategies for holding and transit signal priority. Moses (2005) proposed a simulation model for a CTA bus route. However, he failed to validate his model. According to Moses, validation was not accomplished because of correlations between the parameters, such as the level of passenger demands and specific actions of an operator, who may, for example, decide to speed up [73].

Chen and Chen (2009) [74] conducted a study to identify a way to reduce the waiting time at bus stops. The author used time headway adherence to analyze and measure the reliability of bus services on high-frequency routes. According to this simulation model, more significant variations in running time and in the arrival of passengers at the station results in greater headway variation and increase the average waiting time. Greater variations in running time also increase cumulative headway variations and increase the unreliability of the service.

Altun and Furth (2009) [75] employed Monte Carlo and traffic microsimulation models to examine the prioritization of transit signals. Using these models, the authors detected delays at dispatch and traffic signals and assessed the impact of crowding on dwell time. They also examined different strategies for operational control and signal priority. Delgado et al. (2009) [76] assessed the efficiency of control actions performed in real-time, such as placing limits on holding and boarding. The goal of their study was to determine how to enhance headway

regularity on a hypothetical bus line. The authors used both simulation and optimization to identify a way to enhance bus service. Larrain et al. (2010) [77] developed an optimization framework for the detection of the appropriate configuration of express services offered on a bus route with restricted capacity. The authors simulated a hypothetical route to demonstrate how the model worked and to identify the factors that are most responsible for high-quality service.

Liao et al. (2011) [78] constructed a model for dwell time at certain points, taking into consideration factors such as the time that a bus needs to move between two timing points. This model also included a simulation tool to check the dwell and running times. Using this model, a transport planner can calculate how modifications, including restricted service or consolidation of stops, may impact a route.

According to [79], simulation models are equally applicable to transit operations and traffic assignments. This finding is valid for microscopic models that are used for the simulation of operations close to bus stops, such as those that are used to evaluate the design of stops. Cats (2011) [79] also developed BusMezzo, which is an addition to Mezzo, a mesoscopic traffic simulator. BusMezzo is a simulation model that works at the level of a network. Its focus is on modelling passengers' demand and transit assignments. BusMezzo can be used to evaluate holding strategies and provide information in real-time.

Sanchez-Martinez (2012) [80] proposed a simulation model related to one high-frequency bus line in London, intending to evaluate the distribution of resources along the line. The main contribution of the model was the assessment of the running time distribution using two variables. Running times on different parts of the route were drawn from a distribution of randomly detected bus running times. The detected bus running times were considered to depend on two main factors: the time of day and the running time of a bus in the previous part of the route.

## Gaps in literature

1) According to the review of currently available literature, there is no consistency in reliability definition and indicators. Companies have their own definition of bus service reliability, and they mostly neglect the passengers' perspective of reliability. It can be concluded that there are three main differences between the companies and passengers' perspectives of reliability:

i. Operational reliability is based on timetable adherence, whereas passenger reliability is based on travel time variability.

ii. Operational reliability is usually measured at the route or line-level, whereas passenger reliability is experienced at the origin-destination pair level.

iii. Passenger reliability is experienced for the entire journey, but operational measures capture only portions of the journey.

Accordingly, four different reliability indicators were selected in this study to cover both passengers' and operators' perceptions of reliability: waiting time and on-board crowding level from the passengers' perspective, and headway regularity index (on-time performance) and bus bunching/big gap percentage from operators' perspective.

2) The findings of this review demonstrate that agencies and researchers need to pay particular attention to passengers' perceptions before and after the implementation of the improvement strategies to increase bus service reliability. For example, several studies have focused on the impact of such strategies on waiting time, but it is rare to find evaluations of passengers' perceptions before and after the implementation of such strategies. The simulation model is a

useful tool that helps researchers to capture the effect of the strategies after implementation. For instance, the simulation would calculate and report the waiting time and excess waiting time after the implementation of the strategies.

3) Furthermore, a few studies have been conducted to investigate the impact of the component of two or more strategies on service reliability. This knowledge is necessary because, in the real world, most bus providers implement a component of strategies to maintain regular and reliable service. Therefore, the combinations of the selected strategies were also implemented on the route, and the results were captured and compared.

The primary objective of this research is to improve the reliability of high-frequency bus service and simulation tools currently used in the public transportation companies. The collected studies analyzed bus service reliability using different methodologies. However, to the best of our knowledge, there is virtually no study that has considered all these three gaps comprehensively. Therefore, we proposed a bus service simulation model that covers three main gaps 1) evaluating the bus service reliability from both passengers' and agencies' perspective, 2) considering passenger behavior before and after implementation of strategies and 3) being able to implement and analyze the impact of the component of strategies.

## Simulation model conceptual framework

A flexible methodology for developing public transport simulation model is presented in this section. Although it has been developed using only a single route's simulation, its results can be reproduced for other transit routes like the rail systems as well. This bus route's simulation model will be utilized for the implementation of several strategies that are based on the framework. This route includes a list of locations, which all have a specific location controller and a set of distributions. For instance, these distributions can be utilized for modeling extra times & segment running time towards the route's termination. Bus services have two principal locations–bus stops and terminals. To run this simulation successfully, the location controllers demand representation, vehicles, running time distributions, route specifications, and other controller-based parameters must be specified properly. It needs to be followed by replication runs that have an initialization & data collection phase. In the end, observations from all these replications are to be compared together to obtain performance measures. This simulation algorithm has been depicted in an activity diagram in Fig 2.

## Data

RapidKL is a company owned by another government-owned company, Prasarana Berhad, which was formed in 2004 with the objective of providing a solution for the public transport services of Kuala Lumpur and the surrounding cities. In order to monitor and analyze the bus service, RapidKL uses the automatic data collection systems such as Automatic Vehicle Location (AVL), Automatic Passenger Counting (APC) and Automatic Fare Collection (AFC) systems. Primary data for conducting this study was the raw data extracted from these three automatic data sets archive. All the data used in this study were collected for the period of July, August and September 2019 (Only weekdays). Integrating and aggregating these data sets to evaluate a smart transport system is one of the main challenges that transport researchers and agencies currently face [81]. These data sets can be very useful information for other authors and researcher to conduct further studies. Therefore, we decided to provide the cleaned data as an appendix to this paper in both excel and R formats. Raw data is very rich "Big Data" set, which could not be uploaded. However, raw data can be provided upon academic researchers' request.

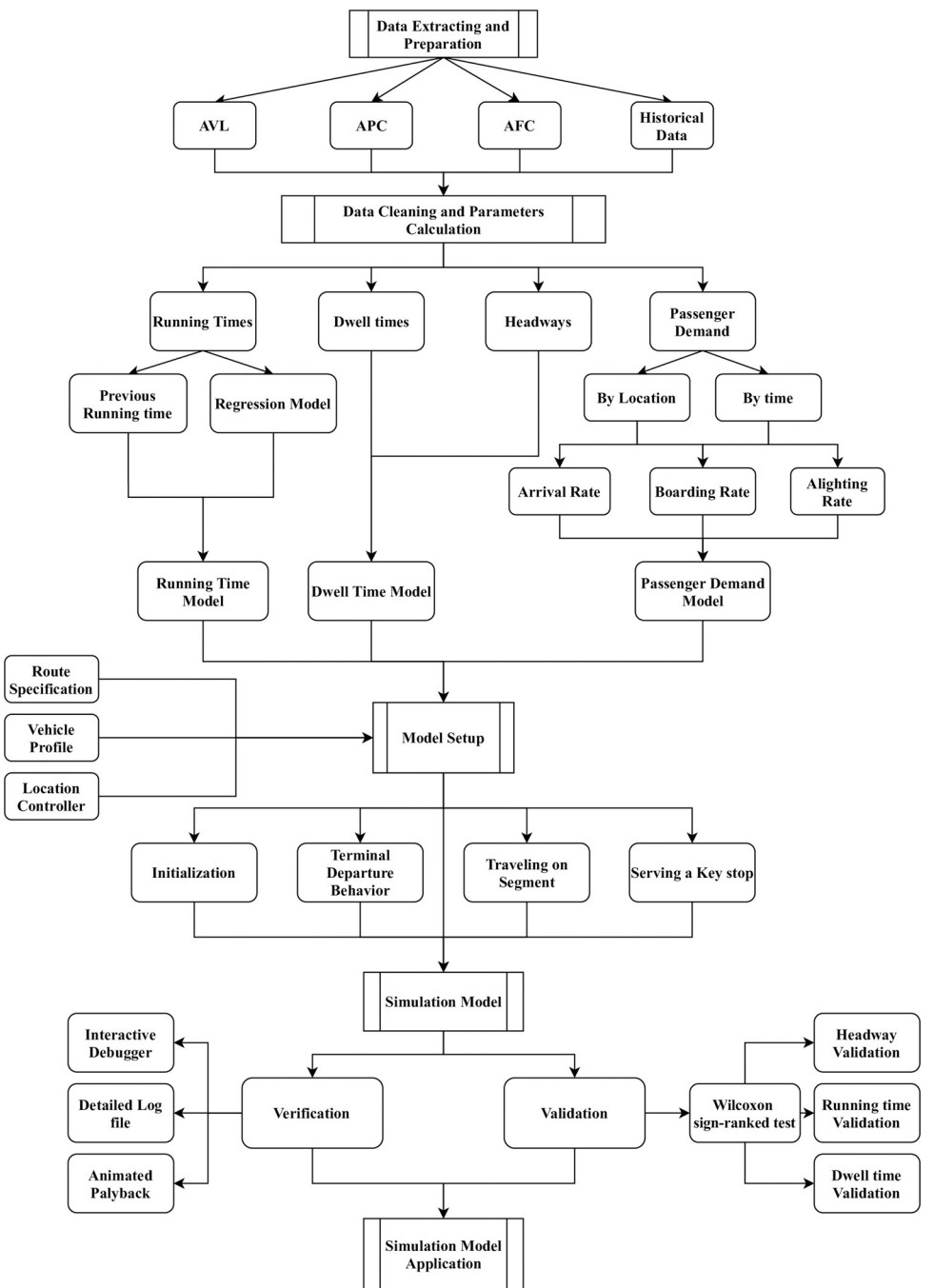

**Fig 2. Simulation model activity diagram showing inputs, outputs, and high-level tasks.**

Route U32 was selected in order to conduct this study. This route passes across the most congested sections of the Kuala Lumpur City Centre (KLCC), providing a suitable environment for conducting a study on service regularity and reliability. Route U32 is a high-frequency route with high passenger demand. This route has fifty-nine bus stops (almost 30 stops in each direction) and nine buses are operating along this route in an operating day. Table 2 demonstrates Route U32 specifications as well as the list of key stops in each direction.

**Table 2. Route U32 key stops.**

| Stop ID | Key-Stop Name | Order | Zone | Distance (meter) |
|---------|---------------|-------|------|------------------|
| **WestBound** | | | | |
| 1000970 | HUB TMN DAGANG | 1 | 3 | 0 |
| 1000360 | BLTN KG PANDAN | 21 | 3 | 5347 |
| 1001846 | MAJESTIC/LRT PUDU | 24 | 3 | 7561 |
| 1000958 | HSBC/7 ELEVEN | 28 | 2 | 9375 |
| **EastBound** | | | | |
| 1000958 | HSBC/7 ELEVEN | 28 | 2 | 0 |
| 1001847 | MAJESTIC/LRT PUDU | 37 | 1 | 3894 |
| 1000359 | BLTN KG PANDAN | 41 | 1 | 5484 |
| 1000970 | HUB TMN DAGANG | 60 | 1 | 11190 |

## Models specification

**Passenger demand model.**   This section explains how passenger demand on a bus route was calculated and modeled across time and by location. Passenger demand is an important component in modeling bus service because it is a primary influence on the dwell time and is necessary to evaluate the impact on the passengers of service variability. Passenger demand is represented as a boarding rate for the entire route, which is then used to distribute boarding and alighting across the key stops and segments of the route in each direction. The day to day variations in passenger boarding were initially analyzed. Once these variations were under control, the variations in passenger boarding was analyzed by time and location along the route. Finally, passenger arrival behavior was analyzed. The result of this analysis formed the model of the passenger demand by day, 15-minute time period, and key stop or segment.

The summary statistics presented in Table 3 represent the day to day variation in passenger demand. There is a range of almost 2500 passengers between the maximum and minimum passenger boarding. Only the passenger data during the weekdays is analyzed. Hence, the overall trip patterns are similar, and the variation in demand is likely due to weather condition (since Malaysia has precarious weather condition and rain can appear at any time of the day).

Fig 3 shows the systematic variation in passenger demand by the hour from the RapidKL estimated weekday boarding data.

The histogram plots in Fig 4 show the passenger boarding of the total period of passenger demand for each period of 15 minutes.

Analyzing the allocation of demand across the key stops and segments of the route is very important in order to determine passenger load profile. Passenger profile is a key component to studying bus running time variation and evaluating service reliability. The distribution of passengers across the route is determined by calculating the aggregate share of passengers served at each key stop and segment according to the direction from the AFC records. Tables 4 and 5 illustrate the summary statistics of passenger activity (boarding + alighting) of Route U32. In order to conduct a useful simulation of a bus route, both key stops and segments boardings (ons) and alightings (offs) are needed.

It is assumed that passengers have arrived key stops after last bus departed. Therefore, we only consider trips which the preceding scheduled trip is recorded in order to analyze the

**Table 3. Route U32 overall passenger demand.**

| | Observation | Min | Max | Mean | Stdev |
|---|---|---|---|---|---|
| **Passenger activity** | 60 | 5438 | 7850 | 6643 | 782 |

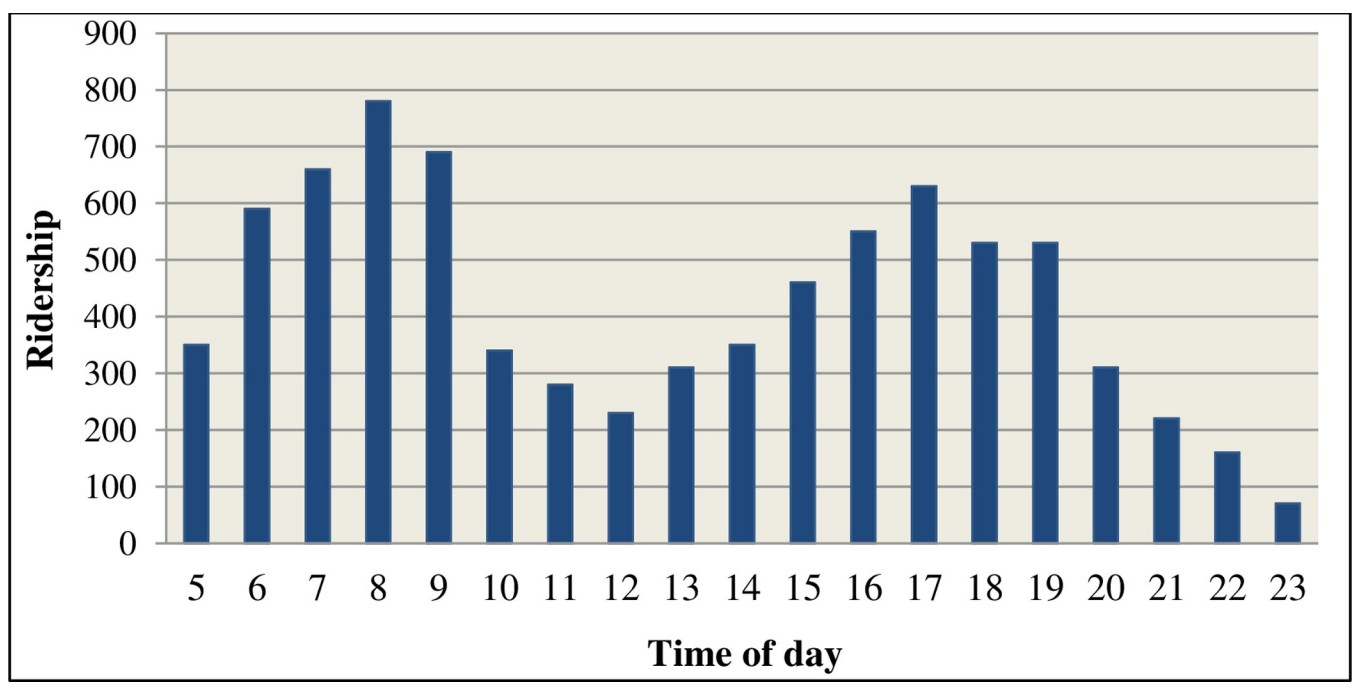

**Fig 3. Route U32 average ridership by time of day.**

arrival rate. This data collection method prevents underestimation of the passenger arrival rate because of an unrecorded trip may in fact be a bus serving passengers. The summary statistics of the passenger arrival rate per minute are shown in Tables 6 and 7.

**Running time model.** The main objective of the modeling the running time was to measure the running time variability and service irregularity at the macroscopic and microscopic level of analysis. Moreover, the final model can be used in the simulation model as the running time generator. The Running Time model developed by Moosavi and Yuen (2020) and Moosavi et al. 2017 was used for analyzing and generating running times [82] [83].

Woodhull (1987) [84] categorized the causes of the unreliable service as being external (exogenous) or internal (endogenous) to the system. Endogenous causes include factors such as driver conduct, poor scheduling, route arrangement, changeable ridership, and inter-bus impacts. Exogenous causes include factors such as traffic incidents and traffic jams, traffic signalization, and intrusions related to on-street parking. Considering the effect of exogenous factors such as weather and traffic jam on the running time variation can be very complicated, and most of the researchers neglected these factors because of lack of enough data and evidence. Since a clear record related to weather or traffic condition was not available for Route U32, the impact of exogenous factors can be tested using the proxy variable of the "running time of the previous bus" (PRT) on the same segment. If there was a significant difference between the running time of the previous bus and the average running time on the same segment, due to, for example, the weather or traffic incidents, then the following trip might also be impacted, if the incident persists long enough to affect both trips. Any effects that do not affect (at least) two successive trips are assumed to be best represented by a random error term in the running time model.

To explore this effect, a subset of the data with a Previous Running Time (PRT) that is significantly different from the average value of running times is examined for correlation between the previously completed trip running time on a segment and the next trip's running time of the same segment. A significant difference in running time is defined as greater than

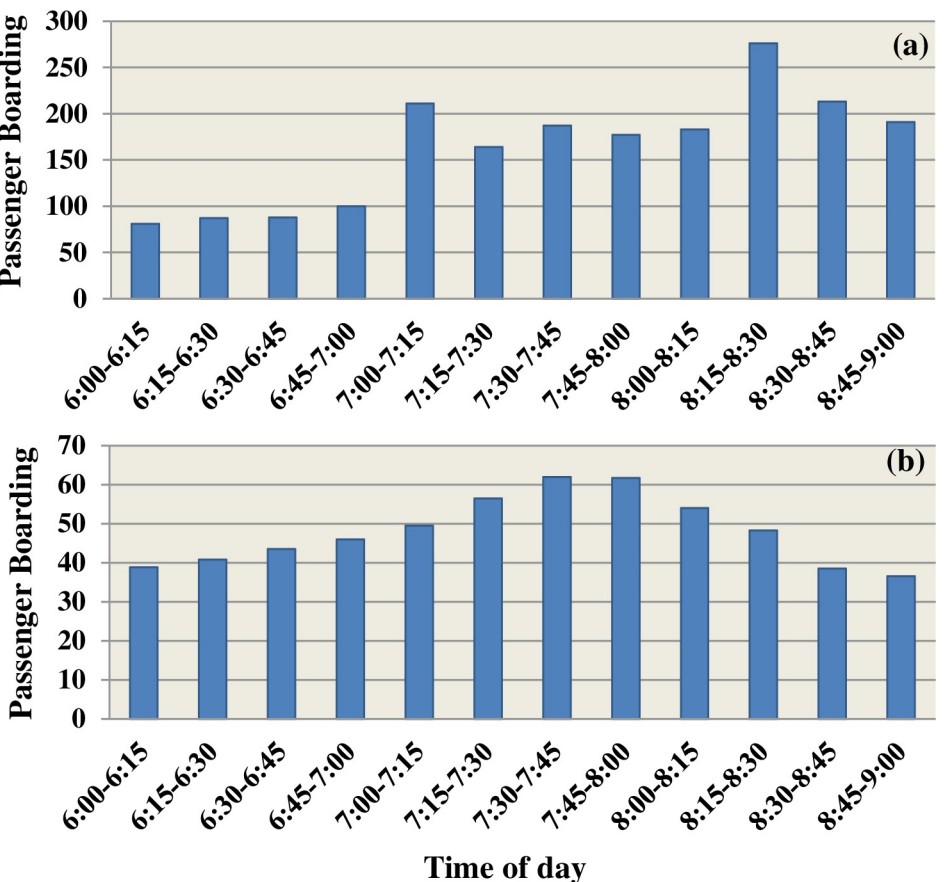

**Fig 4.** Mean of passenger boarding per 15-minute time interval, a) AM peak, out-bound; b) AM peak, in-bound.

the mean plus one standard deviation of the running time. Results of an ordinary least squares regression are displayed in Table 8.

According to Table 8, there is no significant correlation with PRT was observed along the segments. Even constant factor is not significant in all three segments, due to PRT factor. Therefore, external factors such as weather and traffic situation do not appear to impact bus running times consistency in this case. Maybe variations in drivers' behavior control the correlation of the travel time on successive trips. For more information on running time modeling and factors effecting running time, please refer to [83].

**Dwell time model.** Bus service reliability can be impacted directly by dwell time. Dwell time itself can be affected by various factors such as passenger activity, holding and operator relief. To date, a number of researches were conducted by transit companies and researchers to consider and evaluate other functions of the dwell time. A comprehensive descriptive

**Table 4. Route U32 passenger activity statistics summary; AM peak, outbound.**

|  |  | Key stop Ons | | | Key stop Offs | | | Segment Ons | | | Segment Offs | | |
|---|---|---|---|---|---|---|---|---|---|---|---|---|---|
|  | Observed | Mean | Max | St.dev | Mean | Max | St.dev | Mean | Max | St.dev | Mean | Max | St.dev |
| PANDAN | 200 | 24.1 | 34 | 7.6 | 6 | 8 | 1.8 | 36 | 47 | 7.3 | 4 | 8 | 2.1 |
| MAJESTIC | 200 | 11.4 | 25 | 6.4 | 11.7 | 15 | 3.2 | 10.2 | 15 | 3.4 | 15.1 | 21 | 4.4 |
| HSBC | 200 | 6.7 | 9 | 2.1 | 22.2 | 31 | 5.1 | 3.5 | 6 | 2.1 | 23 | 33 | 6.7 |

**Table 5. Route U32 passenger activity statistics summary; AM peak, in-bound.**

| | | Key stop Ons | | | Key stop Offs | | | Segment Ons | | | Segment Offs | | |
|---|---|---|---|---|---|---|---|---|---|---|---|---|---|
| | Observed | Mean | Max | St.dev | Mean | Max | St.dev | Mean | Max | St.dev | Mean | Max | St.dev |
| MAJESTIC | 200 | 5.3 | 12 | 3.6 | 5.3 | 10 | 3.1 | 21.6 | 29 | 4.1 | 6.8 | 11 | 3.8 |
| PANDAN | 200 | 3.1 | 7 | 2.6 | 5.7 | 14 | 4.2 | 6.4 | 10 | 2.1 | 9.3 | 15 | 3.2 |
| HUB | 200 | 0.0 | 0.0 | 0.0 | 3.2 | 6 | 2.3 | 7.2 | 9 | 2.1 | 10.3 | 16 | 4.4 |

analysis and regression model on the factors affecting dwell time, such as passenger activity lift operation and crowding was previously conducted by Moosavi et al. (2017) [85]. In this study, we used this study on dwell time model, in order to evaluate and generate dwell times in our simulation model.

**Model setup.** Each vehicle has its own profile which indicates the time and location of removal and insertion. With each step up, a new vehicle gets introduced, whereas each step-down will result in removal. This removal takes place at the terminals where the first vehicle arrives after the vehicle profile undergoes a decrement. Passenger arrivals and departures at alighting, boarding, and stops are included in this model. The boarding rates and other data at the stops are utilized for aggregating performance measures in order to weigh the higher ridership stops more heavily. As shown in Fig 5, each replication consists of four phases: initialization, serving a key stop, serving segment and terminal recovery.

**Verification and validation.** The verification & validation steps are highly fundamental during the modeling stage. It is clearly stated by North (2007):*"Before verification and validation, models are toys; after verification and validation, models are tools"* [86]. Various verification tests that were conducted on the simulation algorithm and individual components did not show any errors. When it comes to simulation modeling, it is verification that ensures that the codes are working as intended [87]. The verification itself consists of three steps: 1- running the codes line by line to find any bug and invalid values. 2- the simulation's log file contains a detailed record of decisions and actions made during the program's execution and 3- animated playback depicts various vehicles and their daily movements at a highly accelerated pace. Fig 6 illustrates a specific snapshot of this animated playback.

The morning peak hour period was selected for testing and validating the simulation model, since this period has the highest passenger demand, and the impact of strategies on service reliability would be most significant.

The simulation was validated by comparing the running time, headway, and dwell time, which were calculated by plugging real AFC/AVL data into the simulation. Wilcoxon signed-ranking test was used in order to compare real-world situation with the simulation models' out-puts. According to the validation tests' results, the estimated dwell times, headways and running times are valid with relatively small errors, especially in terms of dwell times and running times. Estimated headways show a higher percentage of rejection compared to the dwell and running times. Although the verification tests are successful and the code is extensively revised where needed, it was likely that algorithm programming or design errors caused these

**Table 6. Passenger arrival rate per minute for Route U32, AM peak, outbound.**

| | | Passengers per Minute | | |
|---|---|---|---|---|
| | Observation | Mean | St.dev | Max |
| HSBC | 200 | 2.0 | 1.7 | 4.3 |
| MAJESTIC | 200 | 1.2 | 0.7 | 2.8 |
| PANDAN | 200 | 0.8 | 0.7 | 2.7 |

**Table 7. Passenger arrival rate per minute for Route U32, AM peak, inbound.**

| | Observation | Passengers per Minute | | |
|---|---|---|---|---|
| | | Mean | St.dev | Max |
| **PANDAN** | 200 | 3.1 | 1.1 | 5.2 |
| **MAJESTIC** | 200 | 1.8 | 0.9 | 3.2 |
| **HSBC** | 200 | 2.1 | 0.9 | 4 |

differences. However, the difference was not significant. Table 9 presents a summary of the Wilcoxon-sign-ranked test after 1000 run for each key stop or segments.

## Measuring bus service reliability

As mentioned earlier in the review of currently available literature, four indicators were selected to evaluate bus service reliability: Headway Regularity Index at Key Stops (HRIS), Waiting Time, Big-gap/bunching and onboard crowding level. Headway variability is probably a more direct measurement of transit service reliability. This is followed by passengers' level of service, then travel time variability since the headways determines how long, on average, passengers wait for a bus.

Short headway and high passenger demand are the two most significant characteristics of high-frequency service. In addition, passengers tend to arrive at stops more randomly instead of rigidly following the timetable. In such situation, reliability can be measured by the service providers' ability to minimize headway variations and average waiting time for the passengers. Accordingly, the headway regularity index at stops (HRIS) is designed to measure headway reliability at one specific point. The equation below can be used to calculate HRIS. When HRIS is equal to zero, there is no variation in the headways, and when the value is equal to 1.0, it indicates that bus bunching or big gaps happen frequently.

$$HRIS = \frac{\sum_i \left| \frac{H_{i,j} - H'_{i,j}}{H_{i,j}} \right|}{n} \tag{1}$$

Where:

HRIS: headway regularity index based on stops.

$H_{i,j}$ = Scheduled headway for bus i at stop j.

$H'_{i,j}$ = Actual headway for bus i at stop j.

n = number of buses serve stop j.

Bus bunching and big gap were indicated as causes and consequences of bus service unreliability. Both factors were introduced as the cause and the consequence of unreliability. Headways of less than 60 would be considered as bunching and headways more than twice the related scheduled headway would be considered as big gap.

The scheduled waiting time (expected waiting time) is an important indicator of on-time performance for both agencies and passengers on high-frequency routes, where passengers

**Table 8. Route U32 previous running time correlations.**

| Segment | Observed | PRT | t-stat | constant | t-stat | AdjR$^2$ |
|---|---|---|---|---|---|---|
| **Segment 1** | 72 | 0.65 | 1.68 | 103 | 0.68 | 0.45 |
| **Segment 2** | 43 | 0.88 | 5.1 | 331 | 3.45 | 0.21 |
| **Segment 3** | 51 | 0.48 | 1.79 | 239 | 1.53 | 0.41 |

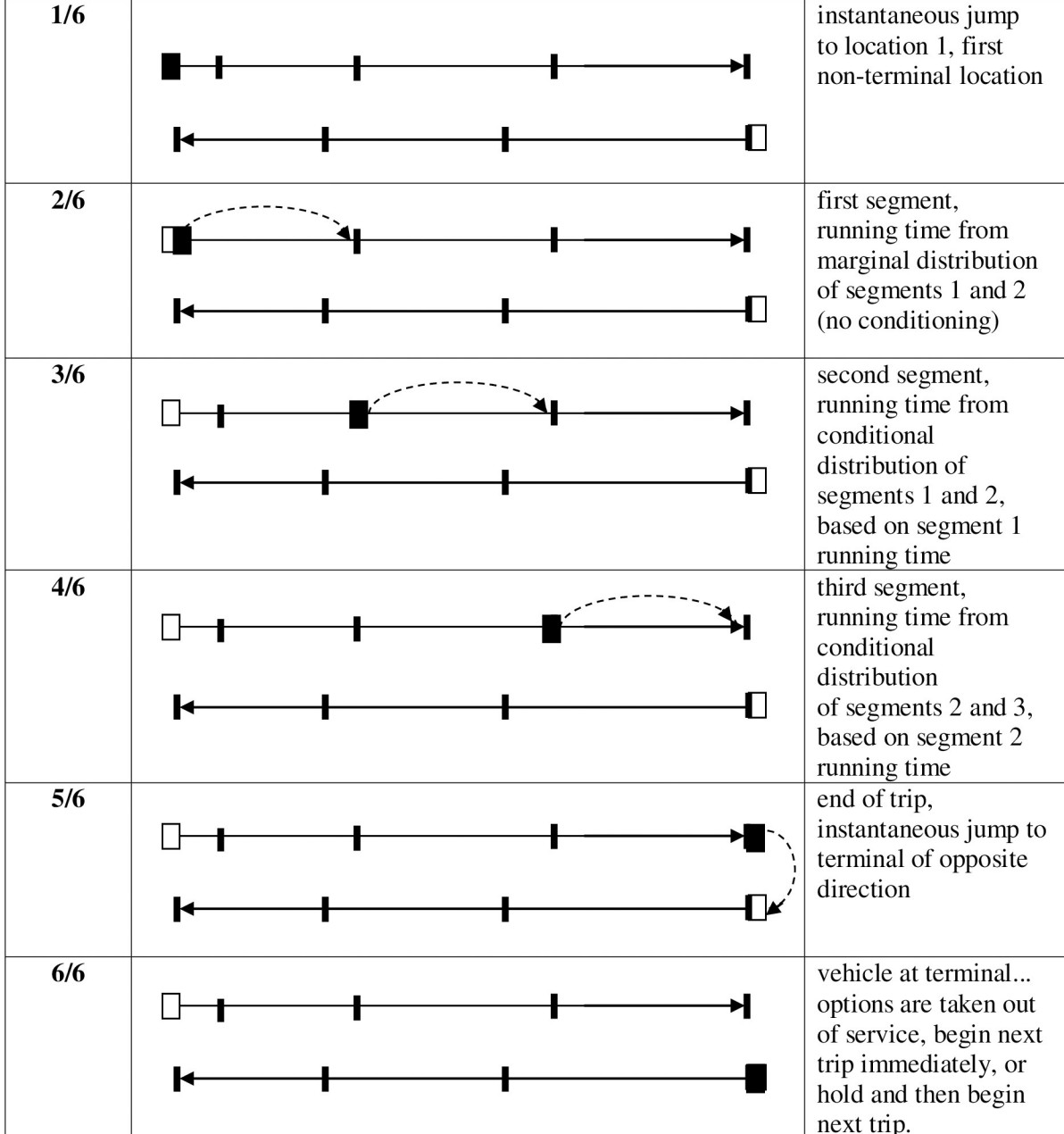

| 1/6 | | instantaneous jump to location 1, first non-terminal location |
|-----|---|---|
| 2/6 | | first segment, running time from marginal distribution of segments 1 and 2 (no conditioning) |
| 3/6 | | second segment, running time from conditional distribution of segments 1 and 2, based on segment 1 running time |
| 4/6 | | third segment, running time from conditional distribution of segments 2 and 3, based on segment 2 running time |
| 5/6 | | end of trip, instantaneous jump to terminal of opposite direction |
| 6/6 | | vehicle at terminal... options are taken out of service, begin next trip immediately, or hold and then begin next trip. |

**Fig 5. Walk-through of events for a single vehicle.**

arrive randomly at the stop without considering time table. Expected waiting time for a route with perfectly regular headway (no headway deviation) is half of the scheduled headway. The exact value of the expected waiting time in the presence of headway variations can be calculated by Eq 2.

$$E(w) = \frac{E(h)}{2}[1 + Cov^2(h)] \qquad (2)$$

Where [h] is the average headway, and cov(h) is the coefficient of variation of headways.

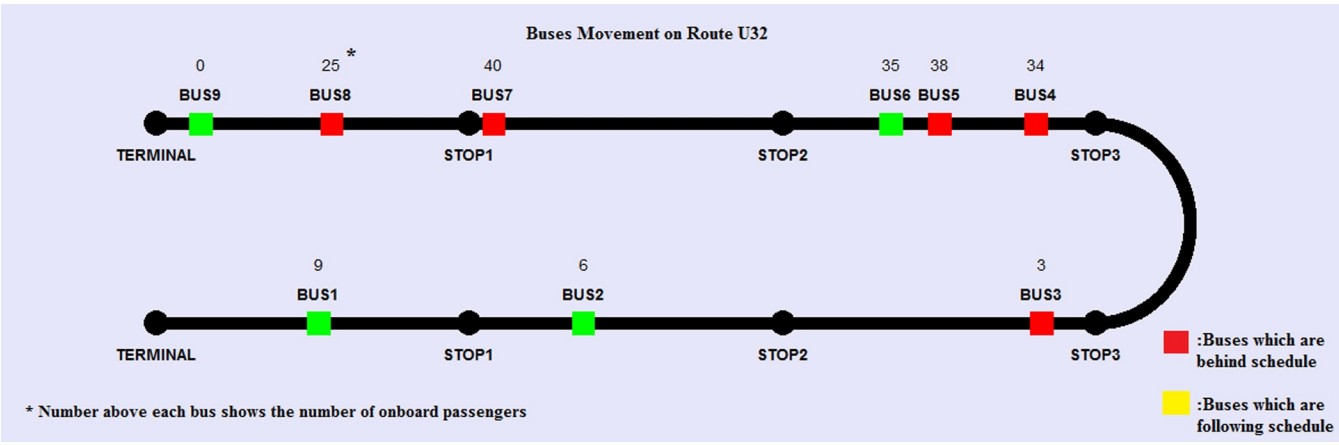

**Fig 6. Snapshot of animated playback for verification propose.**

Another measure of passenger wait time for high-frequency bus routes is the passenger *excess waiting time* (EWT). Excess waiting time is simply the difference between the actual waiting time and the scheduled waiting time. Therefore, excess waiting time is an appropriate indicator which shows service headway unreliability in terms of the passengers' waiting time. An average Excess Wait Time is then calculated for each key stop and different headway pattern using the equation below:

$$EWT = AWT - SWT \tag{3}$$

Eq 4 shows an extended version of Eq 3. Excess Waiting Time is equal to Actual minus Schedule Waiting Time. The Actual Waiting Time (AWT) and the Scheduled Waiting Time (SWT) are both calculated using Eq 2 for the average waiting time [*w*]. The only difference is in the headways. For calculating AWT, the actual headways which were recorded and extracted from AVL data should be used. While in SWT, the scheduled headways (according to service providers' timetable) should be used.

$$EWT = \frac{1}{2}E[h_{actual}](1 + \text{cov}^2(h_{actual})) - \frac{1}{2}E[h_{schedule}](1 + \text{cov}^2(h_{schedule})) \tag{4}$$

## Policy and strategy intervention by simulation model

In this section, the results obtained from the model are presented after different control strategies have been tested. Improving transit service quality is not possible without proper control strategy and policy. Transit companies control the quality of the services using automatic data collection systems and continuous monitoring to provide the most reliable service. As explained before,

**Table 9. Validation summary (Wilcoxon sign-ranked test).**

| Location | Dwell time reject (%) | Headway reject (%) | Running time reject (%) |
|---|---|---|---|
| Key stop 1 | 0.3 | 14.7 | – |
| Key stop 2 | 0.9 | 7.7 | – |
| Key stop 3 | 0.2 | 7.8 | – |
| Segment 1 | – | – | 2.8 |
| Segment 2 | – | – | 2.8 |
| Segment 3 | – | – | 4.1 |

**Table 10. Effect of terminal dispatch strategies on waiting time.**

| Strategy | W.T (sec) | E.W.T (sec) | % of Sch. W.T (300 s) | W.T change (%) | E.W.T change (%) |
|---|---|---|---|---|---|
| Headway-based dispatch | 417 | 117 | 139% | -51.22 | -78.91 |
| Schedule-based dispatch | 570 | 270 | 190% | -33.18 | -51.17 |

four different types of strategies are selected and implemented based on the specifications of Route U32 (Kuala Lumpur) as shown in Table 1. In order to fulfill the gap in the literature and cover both passenger and agencies' perspective, the reliability of bus service is evaluated based on headway regularity index (HRIS), bus bunching/big gap percentage, passenger waiting time (excess wait time) and passenger crowding reliability metric. Strategies were implemented on the route at the terminal and key stops to find the optimum impact on bus service reliability.

The peak morning hours of the operation days were simulated, from the beginning to the end (from early hours to 9:00 AM). Therefore, there are no boundary conditions to be specified. It is possible to run a simulation of the operations with specific boundary conditions. However, the best way to warm the model up is by using simulations from the day's start to the selected period.

## Dispatching strategies at terminals

As for the management strategy, terminal recovery times (or time till the next departure) for buses have been calculated as the greater of the scheduled headway minus the previous headway and zero. Previous headway refers to the time at which the previous bus departed from the terminal. To put it simply, buses are instructed to depart their terminal at approximately the time of their scheduled headway. Tables 4–6 present the impact of headway-based dispatching strategy on bus service reliability indicators.

According to the results of the terminal dispatching strategies (Tables 10–12), headway-based departures showed considerably better results, especially in terms of headway regularity indexes. As long as Route U32 is a high-frequency route, it was expected that headway-based strategies should have more impact on reliability improvement, compared to schedule-based strategies.

Schedule-based departure improves both waiting time (WT) and bunching/big gaps significantly, but no significant impact on headway regularity is observed. On the other hand, headway-based dispatch policies have a highly significant impact on headways, as presented in Table 11. Negative percentage indicates the reduction in that specific indicator, after implementing strategy. For example, according to Table 12, big gap is decreased by 90% after implementing Headway-based dispatching strategy. Moreover, the average waiting time improved significantly by implementing both headway-based and schedule-based dispatching strategies by 51% and 33%, respectively. However, the average waiting times are still higher than the planned waiting time (1 min and 57 seconds). Fig 7 illustrates a comparison between these two terminal dispatching strategies.

## Holding strategies at key stops

Key stops also can be suitable locations for implementing strategies. Bus operators usually use bus stops with the highest boarding rates for adjusting service regularity. Passengers in high-

**Table 11. Effect of terminal dispatch strategies on headway regularity index.**

| Strategy | HRIS 1 Change (%) | HRIS 2 Change (%) | HRIS 3 Change (%) | HRIS 4 Change (%) | HRIS 5 Change (%) |
|---|---|---|---|---|---|
| Headway-based dispatch | 0.20 (-74%) | 0.32 (-72%) | 0.52 (-53%) | 0.62 (-54%) | 0.64 (-58%) |
| Schedule-based dispatch | 0.71 (-7.8%) | 1.144 (-1.3) | 0.95 (-16%) | 1.347 (-1.5%) | 1.643 (-0.64%) |

**Table 12. Effect of terminal dispatch strategies on big gap and bunching.**

| Strategy | Big gap (%) | Bunching (%) | Big gap change (%) | Bunching change (%) |
|---|---|---|---|---|
| Headway-based dispatch | 2.00 | 0.0 | -90 | -100 |
| Schedule-based dispatch | 3.5 | 0.0 | -83.33 | -100 |

frequency bus routes do not pay attention to schedule and usually arrive at stops randomly. Therefore, regular bus service is much more important than adherence to schedule in high-frequency routes. Accordingly, only headway-based strategies were selected for implementation at the key stops. Previous headway and Prefol Headway strategies were tested on Route U32, and the results are presented through Tables 13–15.

Both strategies have shown a similar impact on the reliability indicators. Waiting time is reduced by 44% and 43% after the previous, and Prefol Headway strategies are adopted, respectively. Terminal dispatching strategies have better results in terms of waiting time because by implementing holding strategies at key stops, on-board passengers could be held until departure time. One of the added benefits of implementing the strategies at the terminal is that no passenger would be affected by the extra time when the holding strategy is implemented. Headway regularity indexes are improved at all the key stops by almost 50%. Although it is not exactly the ideal regularity, 50% to 60% improvement is an acceptable value. There are some factors affecting bus service regularity that are totally out of the operators' controls such as the weather, traffic flow, vehicle break downs and accidents along the route. These factors can decrease regularity, even in routes under strict control. Fig 8 compares the impact of previous and Prefol Headway strategies at key stops on reliability indicators.

## Combination of strategies

Four different types of strategies were tested on route U32 (two dispatching strategies at terminals and two holding strategies at key stops). All the adopted strategies improved bus service reliability significantly. It is also possible that operators adopt a set of strategies in order to improve various aspects of reliability. But before taking any action, bus companies need a clear understanding of which set of strategy have what effect on bus service reliability. This section presents the results of adapting a combination of different strategies on Route U32.

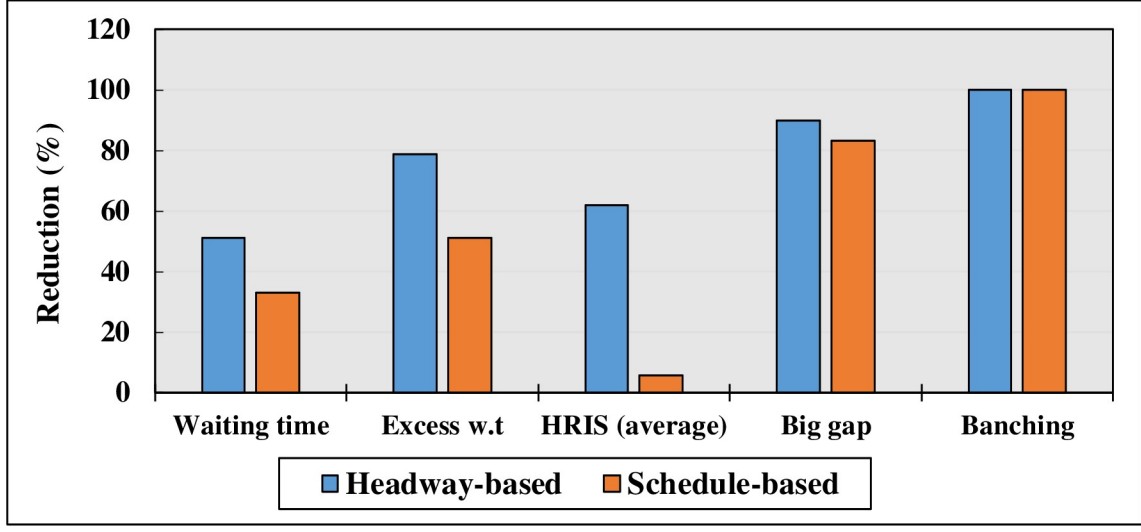

**Fig 7. Comparison of headway-based and schedule-based dispatching strategies.**

**Table 13. Effect of holding strategies on waiting time.**

| Strategy | W.T (sec) | E.W.T (sec) | % of Sch. W.T (300 s) | W.T change (%) | E.W.T change (%) |
|---|---|---|---|---|---|
| Previous headway | 475 | 175 | 158% | -44% | -68% |
| Prefol headway | 479 | 179 | 159% | -43% | -67% |

W.T = waiting time; E.W.T = excess waiting time; Sch W.T = scheduled waiting time.

**Table 14. Effect of holding strategies on big gap and bunching.**

| Strategy | Big gap (%) | Bunching (%) | Big gap change (%) | Bunching change (%) |
|---|---|---|---|---|
| Previous headway | 1 | 0.0 | -95 | -100 |
| Prefol headway | 1 | 0.0 | -95 | -100 |

**Table 15. Effect of holding strategies on headway regularity index.**

| Strategy | HRIS 1 Change (%) | HRIS 2 Change (%) | HRIS 3 Change (%) | HRIS 4 Change (%) | HRIS 5 Change (%) |
|---|---|---|---|---|---|
| Previous headway | 0.40 (-48%) | 0.44 (-62%) | 0.59 (-47%) | 0.67 (-51%) | 0.65 (-58%) |
| Prefol headway | 0.40 (-48%) | 0.44 (-62%) | 0.60 (-46%) | 0.69 (-49%) | 0.68 (-56%) |

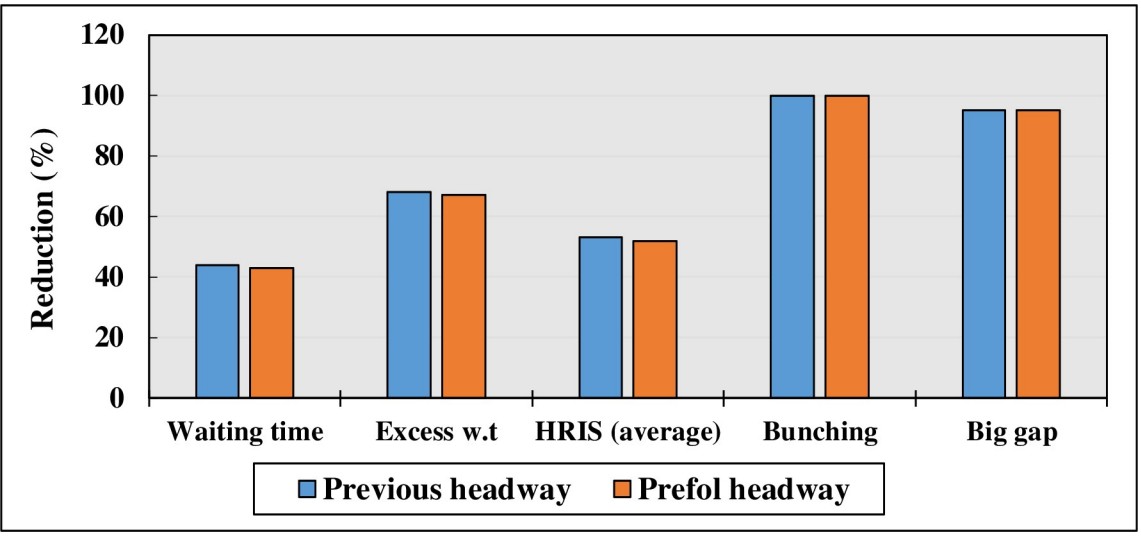

**Fig 8. Comparison of previous and Prefol Headway holding strategies.**

**Table 16. Effect of combination of strategies on waiting time.**

| Strategy | W.T (sec) | E.W.T (sec) | % of Sch. W.T (300 s) | W.T change (%) | E.W.T change (%) |
|---|---|---|---|---|---|
| 1 & 3 | 464 | 164 | 154% | -45.73 | -70.45 |
| 1 & 4 | 469 | 169 | 156% | -45.14 | -69.54 |
| 2 & 3 | 403 | 103 | 134% | -52.86 | -81.44 |
| 2 & 4 | 409 | 109 | 136% | -52.16 | -80.36 |

**Table 17. Effect of combination of strategies on big gap and bunching.**

| Strategy | Big gap (%) | Bunching (%) | Big gap change (%) | Bunching change (%) |
|---|---|---|---|---|
| 1 & 3 | 0.0 | 0.0 | -100 | -100 |
| 1 & 4 | 0.0 | 0.0 | -100 | -100 |
| 2 & 3 | 0.0 | 0.0 | -100 | -100 |
| 2 & 4 | 0.0 | 0.0 | -100 | -100 |

Table 16 shows the impact of a combination of strategies on waiting time. According to the results, a combination of Strategies 2 and 3 (Headway-based Departure from terminal and previous Headway Holding at key stops) show the best impact on bus service reliability in terms of passenger waiting time. Waiting time and excess waiting time are both significantly reduced by 52.86% and 81.44%, respectively. In addition, by adapting a combination of strategies, no bus bunching and big gap are observed after 1000 run of the simulation model (Table 17). The results prove that strategies are way more effective when they are combined and implemented together. Moreover, the combination of Strategies 2 and 3 also shows the highest impact on headway regularity, according to Table 18. For instance, headway regularity improves by 74% at Key stop 1, when Strategy 2 and 3 are implemented on the same route and at the same the time. Headway regularity index equals to 0.2 which indicates that 80% of the buses have regular headways and arrive at the key stop on-time. Fig 9 presents a comparison on the results of the combination of strategies.

According to Fig 9, a combination of strategies 2 and 3 (Headway-based Departure at terminals and previous Headway Holding strategy) can be the most effective strategy for Route U32.

## Conclusion

The main objective of this study was to develop a micro-simulation package in R studio for the design and evaluation of automatic data collection systems application. The motivations for such a tool include the growing need among transit companies to implement big data and new technologies to improve service reliability, compete with the private transport sector, and the ever-present gap between passengers' satisfaction and the level of services offered by bus service providers that have burrowed its way deep into the fabric of bus transit systems.

This study was designed to cover gaps which have been recognized in the literature. To the best of our knowledge, currently, there is no bus service simulation package available to 1) Analyze and measure the level of bus service reliability **considering both passengers' and agencies' points of view**, 2) Implementing corrective strategies and **combinations of strategies** on bus routes to find out the effect of different strategies, 3) Capturing and comparing the level of reliability **before and after** implementing any changes for Route and 4) Analyzing "Headway Regularity" in terms of regularity index, big gap and bunching and excess waiting time.

Headway-based Departure strategy offered the best result compared to the other strategies used. In addition, the combinations of the corrective strategies were implemented on the route

**Table 18. Effect of combination of strategies on headway regularity index.**

| Strategy | HRIS 1 Change (%) | HRIS 2 Change (%) | HRIS 3 Change (%) | HRIS 4 Change (%) | HRIS 5 Change (%) |
|---|---|---|---|---|---|
| **1 & 3** | 0.31 (-59%) | 0.40 (-65%) | 0.58 (-48%) | 0.66 (-51%) | 0.64 (-58%) |
| **1 & 4** | 0.31 (-59%) | 0.40 (-65%) | 0.58 (-48%%) | 0.67 (-51%) | 0.68 (-56%) |
| **2 & 3** | 0.20 (-74%) | 0.32 (-72%) | 0.51 (-54%) | 0.58 (-57%) | 0.57 (-63%) |
| **2 & 4** | 0.20 (-74%) | 0.32 (-72%) | 0.51 (-54%) | 0.60 (-56%) | 0.60 (-61%) |

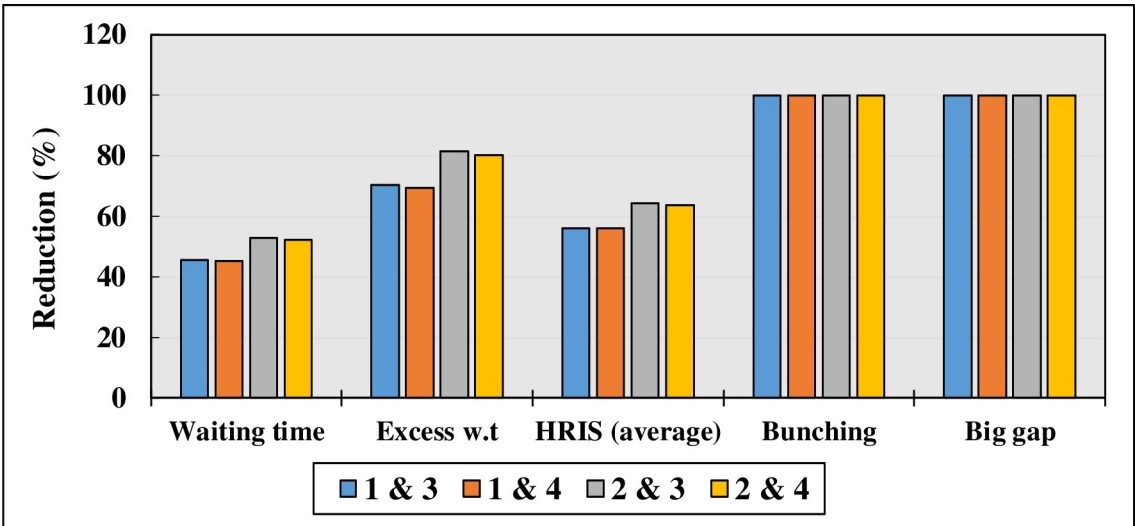

**Fig 9. Comparison of combination of strategies.**

using 1000 simulation runs. Figs 10 and 11 illustrate and compare the summary of the effects of different strategies on the bus service reliability indicators.

According to Fig 10, adapting a combination of Strategy 2 and 3 indicates the best results in terms of reduction in the waiting time. However, Strategy 2 (headway-based dispatching strategy) can reduce the waiting time to almost the same degree as the combination of Strategies 2 and 3. Therefore, a headway-based terminal departure strategy can be the best option for improving the problem of reliability in terms of waiting time.

The results of the strategies test proved that on-time terminal departure could play a critical role in regular bus service. Moreover, the headway-based departure policy improved the level of regularity significantly. Therefore, reducing the terminal departure deviations through a coherent departure strategy would be the most applicable and significant method to reduce

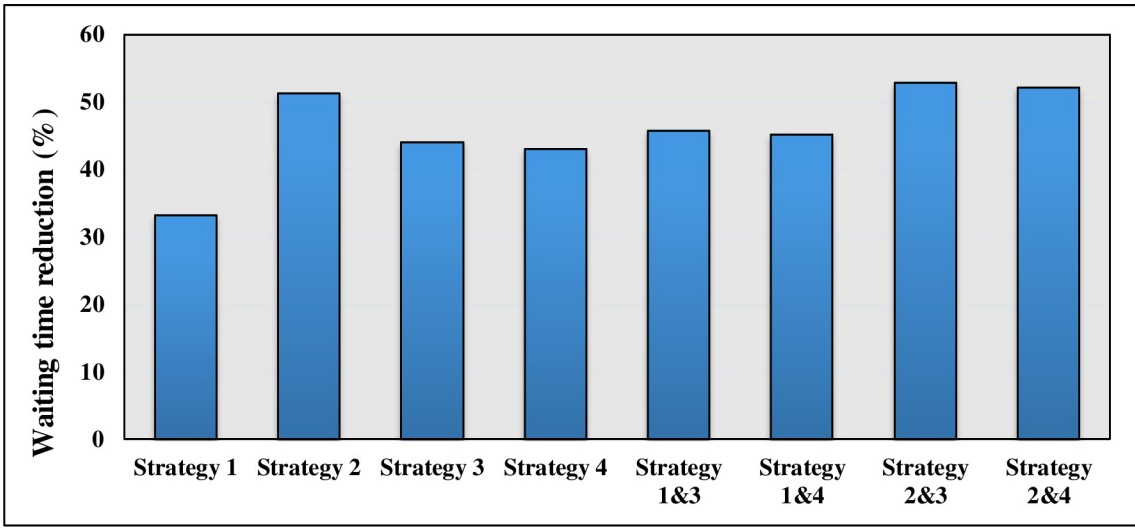

**Fig 10. Comparison of the impact of different strategies on waiting time.**

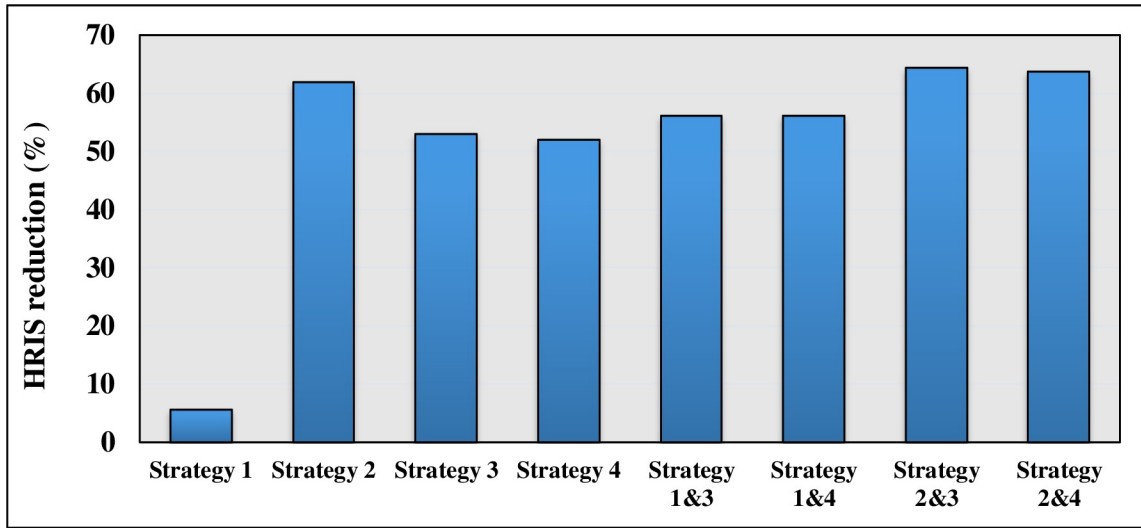

**Fig 11. Comparison of the impact of different strategies on HRIS.**

service unreliability. As long as this strategy only needs to be implemented at the terminal, it would be straightforward for operators and supervisors to adopt this strategy.

All the strategies have almost the same impact on bus bunching and big gaps. Therefore, the comparison of bunching and big gap results will not be presented in the summary. In addition, the effects of the strategies on passenger demands and on-board crowding were not presented in this paper, since the strategies did not show any significant positive effects on the levels of overcrowding. Morning peak hour is the most crowded period of an operational day. In other words, overcrowding is the nature of a high-frequency route in morning peak hours. Accordingly, implementing holding strategies will not decrease the level of crowding significantly.

## Future works

The current version of the simulation model is more focused on analyzing one specific route. Since all simulation codes and Rstudio files are published with this paper, this simulation model can be expanded to evaluate and analyze more than one route at the same time or a network of routes in a specific transport system. Moreover, four different types of control strategies were developed and provided in the simulation model environment for further investigations. However, there are other control strategies, such as short turning and expressing strategies, which can be developed in this simulation model.

## Supporting information

**S1 Dataset.**
(ZIP)

**S2 Dataset.**
(RAR)

## Acknowledgments

The authors would like to acknowledge the Centre for Transportation Research (CTR) of the Faculty of Engineering of the University of Malaya (UM) for providing research facilities.

## Author Contributions

**Conceptualization:** Seyed Mohammad Hossein Moosavi, Amiruddin Ismail.

**Data curation:** Seyed Mohammad Hossein Moosavi, Choon Wah Yuen.

**Funding acquisition:** Choon Wah Yuen.

**Methodology:** Seyed Mohammad Hossein Moosavi.

**Project administration:** Seyed Mohammad Hossein Moosavi, Amiruddin Ismail.

**Software:** Choon Wah Yuen.

**Supervision:** Amiruddin Ismail.

**Validation:** Seyed Mohammad Hossein Moosavi.

**Writing – original draft:** Seyed Mohammad Hossein Moosavi, Choon Wah Yuen.

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
