## [Decision Letter · Decision Letter 0]

2 Mar 2020

PONE-D-20-00591

Using Simulation Model as A Tool for Analysing Bus Service Reliability and Implementing Improvement Strategies

PLOS ONE

Dear Authors,

Thank you for submitting your manuscript to PLOS ONE. After careful consideration, we feel that it has merit but does not fully meet PLOS ONE’s publication criteria as it currently stands. Therefore, we invite you to submit a revised version of the manuscript that addresses the points raised during the review process.

We would appreciate receiving your revised manuscript by 15.4.2020. To enhance the reproducibility of your results, we recommend that if applicable you deposit your laboratory protocols in protocols.io, where a protocol can be assigned its own identifier (DOI) such that it can be cited independently in the future. For instructions see: http://journals.plos.org/plosone/s/submission-guidelines#loc-laboratory-protocols

We look forward to receiving your revised manuscript.

Kind regards,

Dejan Dragan, PhD

Academic Editor

PLOS ONE

Additional Editor Comments (if provided):

Editor's initial comments to the paper:

Using Simulation Model as A Tool for Analysing Bus Service Reliability and

Implementing Improvement Strategies

The primary objective of this research is to improve the reliability of high-frequency bus service and simulation tools currently used in the public transportation companies. Therefore, a simulation model of high-frequency bus service was developed in order to study the strategies to alleviate it. This study was designed to cover gaps that have been recognized in the literature. According to the authors best knowledge, currently, there is no bus service simulation package available in order to 1) Analyse and measure the level of bus service reliability considering both passengers, and agencies point of view, 2) Implementing corrective strategies and combinations of strategies on bus routes to find out the effect of different strategies, 3) Capturing and comparing the level of reliability before and after implementing any changes on the route 4) Illustrating the movement of buses on a specific route and number of onboard passengers in animation playback and 5) Analysing "Headway Regularity" in term of regularity index, big gap and bunching and excess waiting time. Since all simulation codes and Rstudio files are published with this paper, this simulation model can be expanded by other researchers for further studies in the future. The subject of this research is up-to-date and fundamentally interesting for scholars and practitioners from the field. Thus, from this point of view, the paper likely might have deserved an opportunity to be considered for a possible publication. More importantly, we are dealing with an open-source R environment, which is gradually becoming a leading software environment not only for the simulation but for all kinds of different analyses, modeling spheres, and many other tasks.

However, in the current form, the paper, in general, does not satisfy all rigor requirements that are demanded from Plos One. Although a red clue persists more or less consistent all over the paper, the latter suffers from several detected deficiencies. The editor has detected the following issues, which should be corrected prior to continuing a further publishing process:

1. English sentences should be improved.

2. Maybe it would be convenient to add some section or sub-section named “The conceptual framework”, where all consecutive steps of the research should be more clearly emphasized in the form of some block diagram. The latter means a similar structure as it was illustrated in Figure 2, but with more details. Another alternative is to add some pseudo-code clearly demonstrating all research steps and simulation mechanism details.

3. In general, the flow, description, processing, and (intermediate and final) results of all analyses and methods used, might have been, at least in my opinion, better conducted at some places.

4. Please check again if all figures/tables are referenced in the paper.

5. Some figures (e.g., Figure 4) should be improved in the sense of informative content, meaning that the reader immediately understands the main point without even looking at the corresponding text.

Besides the AE comments, here are the comments from the reviewers:

Reviewer 1 (major revision):

The paper deals with the topic of bus service reliability, improvement strategies and simulation model for evaluation of proposed strategies. Authors provided another view on topic, however I disagree with the statement that there is a gap in this field. Other authors and transportation companies just used different measurements to evaluate the service reliability. It would be usefull to compare proposed methodologies and strategies with proposals of other authors. The topic is still up-to-date and the proposed methodology and simulations are interesting, however I have few comments and recommendations for the authors.

- English language in the paper should be corrected.

It contains mistakes and misprints.

- The style of references in the paper should be unified. Authors use various styles.

- Table 1 mentioned on the line 90 does not exist.

Table 1 in the paper contains different information.

- Are the values in the table 2 correct? The length of the line in the westbound direction is twice as long as in the eastbound direction?

- The model from the description in the paper seems to be very simple. It would be usefull to describe the simulation model in more details (describe more the used data, describe the simulation tool used for implementing proposed model, etc.).

- The "cov" in the formula (2) should be described in more details.

- Notation in formula (3) should be described in more details.

- Numerical experiments contain only one bus line. It would be usefull to add a discussion about the influence of other factors on the travelling (more bus lines in the system, interchanges between lines and its influence on waiting times and overall travelling time, etc.)

Reviewer 2 (rejected):

This paper developed a simulation model of bus service to improve reliability of high-frequency bus service. Four different types of strategies (schedulebased departure from terminal, headway-based departure from terminal, previous holding strategy, prefol holding strategy) were selected according to Route U32 (Kuala Lumpur) specifications. It showed that waiting time has been reduced significantly and headway regularity has been improved.

My main concerns are:

1) This paper developed a simulation model, but it does not specifically explain how the random processes are simulated, such as customer's arrival process etc, and how the traffic/other conditions influence the random process. Also, could you provide other parameters in the experiment such as customer size etc.?

2) The paper defined the expected waiting time in presence of headway variation and excess waiting time on page 18, but why defining it this way? What's the intuition behind this definition and why it is a good way to evaluate the service reliability?

3) Following concern 2), the term mean(h_actual) and cov(h_{actual}) are correlated, so how do you calculate EWT specifically in simulation?

4) What's the running time for each simulation test?

5) Usually there would be multiple routes going through one stop, so could this framework/R package be extended to a traffic network?

6) Is there a way to verify the small gap between the simulation result and the real-world implementations?

Reviewer 3 (major revision):

REVIEW REPORT ON PONE-D-20-00591

1. INTRODUCTION:

Lines 52 -73 does not represent an introduction into the subject matter rather it is a brief on what each section of their paper addresses. Author may choose to give a better introductory section on the subject matter then use between 3 to 5 lines to summarize the structuring of their paper.

2. BODY AND RESULT:

Citation style on line 78 should consistent. Why the use of author names and reference number? On line 82, the year of publication for "TCQSM" should be included so as to show currency of thought.

Sentence on line 87 and 88 does not communicate a complete thought.

Generally, the discuss on the subject matter isn't adequate enough to communicate necessary thought by the author.

Line 127 - 130: Author did not include the basis and/or for selection of the four reliability indicators in the study.

Also, lines 133 - 143 refer to previous section of the paper that aren't explicit.

Presentation of reviewed literature needs to be reviewed to convey sequential flow of thought Line 379 also does not give any basis for selection of the four indicators. Author simply referred to literature that isn't cited Tables on lines 454 - 460 seem questionable, probably errors while entering values, otherwise, there is need to justify and/or explain the negative percentage values.

GENERAL COMMENTS:

a) The idea presented by the author is original and seem novel but it is greatly undermined by the author(s)' inability to communicate and accurately present their thoughts.

b) Literature used haven't been properly reviewed to show gap filled by the author's idea.

OVERALL ASSESSESMENT Paper is suitable for publication with PLOS ONE journals BUT NOT in its current state.

According to all reviews, the paper should be likely rejected. However, the Academic Editor does not necessarily share the same opinion. The reason is that the paper, at least from the perspective of the AE, without any doubt, brings certain novelties and contributions. Thus, the AE recommends the authors the following to increase the likelihood of the possible further consideration of the paper: I suggest to strictly follow all the comments of the reviewers and the AE.

Academic Editor DD

Journal Requirements:

"The authors would like to acknowledge the Sustainable Urban Transport Research Centre (SUTRA) of the Faculty of Engineering and Built Environment of the Universiti Kebangsaan Malaysia (UKM) for providing research facilities and the Ministry of Education (MOE) of Malaysia for providing research funding through Project FRGS/2/TK02/UKM/01/1."

"NO - Include this sentence at the end of your statement: The funders had no role in study design, data collection and analysis, decision to publish, or preparation of the manuscript."

Reviewers' comments:

Reviewer's Responses to Questions

**Comments to the Author**

1. Is the manuscript technically sound, and do the data support the conclusions?

Reviewer #1: Partly

Reviewer #2: Partly

Reviewer #3: Partly

2. Has the statistical analysis been performed appropriately and rigorously? 

Reviewer #1: Yes

Reviewer #2: No

Reviewer #3: Yes

3. Have the authors made all data underlying the findings in their manuscript fully available?

Reviewer #1: No

Reviewer #2: Yes

Reviewer #3: Yes

4. Is the manuscript presented in an intelligible fashion and written in standard English?

Reviewer #1: Yes

Reviewer #2: Yes

Reviewer #3: No

5. Review Comments to the Author

Reviewer #1: The paper deals with the topic of bus service reliability, improvement strategies and simulation model for evaluation of proposed strategies. Authors provided another view on topic, however I disagree with the statement that there is a gap in this field. Other authors and transportation companies just used different measurements to evaluate the service reliability. It would be usefull to compare proposed methodologies and strategies with proposals of other authors.

The topic is still up-to-date and the proposed methodology and simulations are interesting, however I have few comments and recommendations for authors.

- English language in the paper should be corrected. It contains mistakes and misprints.

- The style of references in the paper should be unified. Authors use various styles.

- Table 1 mentioned on the line 90 does not exist. Table 1 in the paper contains different information.

- Are the values in the table 2 correct? The length of the line in the westbound direction is twice as long as in the eastbound direction?

- The model from the description in the paper seems to be very simple. It would be usefull to describe the simulation model in more details (describe more the used data, describe the simulation tool used for implementing proposed model, etc.).

- The "cov" in the formula (2) should be described in more details.

- Notation in formula (3) should be described in more details.

- Numerical experiments contain only one bus line. It would be usefull to add a discussion about the influence of other factors on the travelling (more bus lines in the system, interchanges between lines and its influence on waiting times and overall travelling time, etc.)

Reviewer #2: This paper developed a simulation model of bus service to improve reliability of high-frequency bus service. Four different types of strategies (schedule-based departure from terminal, headway-based departure from terminal, previous holding strategy, prefol holding strategy) were selected according to Route U32 (Kuala Lumpur) specifications. It showed that waiting time has been reduced significantly and headway regularity has been improved.

My main concerns are:

1) This paper developed a simulation model, but it does not specifically explain how the random processes are simulated, such as customer's arrival process etc, and how the traffic/other conditions influence the random process. Also, could you provide other parameters in the experiment such as customer size etc.?

2) The paper defined the expected waiting time in presence of headway variation and excess waiting time on page 18, but why defining it this way? What's the intuition behind this definition and why it is a good way to evaluate the service reliability?

3) Following concern 2), the term mean(h_actual) and cov(h_{actual}) are correlated, so how do you calculate EWT specifically in simulation?

4) What's the running time for each simulation test?

5) Usually there would be multiple routes going through one stop, so could this framework/R package be extended to a traffic network?

6) Is there a way to verify the small gap between the simulation result and the real-world implementations?

Reviewer #3: REVIEW REPORT ON PONE-D-20-00591

1. INTRODUCTION:

Lines 52 -73 does not represent an introduction into the subject matter rather it is a brief on what each section of their paper addresses. Author may choose to give a better introductory section on the subject matter then use between 3 to 5 lines to summarize the structuring of their paper.

2. BODY AND RESULT:

Citation style on line 78 should consistent. Why the use of author names and reference number?

On line 82, the year of publication for “TCQSM” should be included so as to show currency of thought.

Sentence on line 87 and 88 does not communicate a complete thought.

Generally, the discuss on the subject matter isn’t adequate enough to communicate necessary thought by the author.

Line 127 - 130: Author did not include the basis and/or for selection of the four reliability indicators in the study.

Also, lines 133 – 143 refer to previous section of the paper that aren’t explicit.

Presentation of reviewed literature needs to be reviewed to convey sequential flow of thought

Line 379 also does not give any basis for selection of the four indicators. Author simply referred to literature that isn’t cited

Tables on lines 454 – 460 seem questionable, probably errors while entering values, otherwise, there is need to justify and/or explain the negative percentage values.

GENERAL COMMENTS:

a) The idea presented by the author is original and seem novel but it is greatly undermined by the author(s)’ inability to communicate and accurately present their thoughts.

b) Literature used haven’t been properly reviewed to show gap filled by the author’s idea.

OVERALL ASSESSESMENT

Paper is suitable for publication with PLOS ONE journals BUT NOT in its current state.

6. PLOS authors have the option to publish the peer review history of their article (what does this mean?). If published, this will include your full peer review and any attached files.

Reviewer #1: No

Reviewer #2: No

Reviewer #3: No

---

## [Author Response · Author response to Decision Letter 0]

13 Apr 2020

Authors’ Response to the Reviewer Comments

Journal: PLOS ONE 

Manuscript ID: PONE-D-20-00591

Title of Paper: Using Simulation Model as a Tool for Analyzing Bus Service Reliability and Implementing Improvement Strategies

Authors: Seyed Mohammad Hossein Moosavi, Amiruddin Ismail , and Choon Wah Yuen

Dear Editor and Reviewers, 

We appreciate the time and efforts by the editor and referees in reviewing this manuscript. We have addressed all issues indicated in the review report, and believed that the revised version can meet the journal publication requirements. As below, on behalf of my co-authors, I would like to clarify some of the points raised by the Reviewers. We hope the Reviewers and the Editors will be satisfied with our responses to the ‘comments’ and the revisions for the original manuscript. 

Thanks and Best Regards!

Sincerely,

Seyed Mohammad Hossein Moosavi (Ph.D)

Centre for Transportation Research (CTR)

Faculty of Engineering

University of Malaya (UM), 

Kuala Lumpur, Malaysia

 April 10, 2020

Comment from editor

1. English sentences should be improved.

Thank you for your suggestion, the manuscript has been reviewed and edited throughout the manuscript by the expert English editor. 

2. Maybe it would be convenient to add some section or sub-section named “The conceptual framework”, where all consecutive steps of the research should be more clearly emphasized in the form of some block diagram. The latter means a similar structure as it was illustrated in Figure 2, but with more details. Another alternative is to add some pseudo-code clearly demonstrating all research steps and simulation mechanism details. In general, the flow, description, processing, and (intermediate and final) results of all analyses and methods used, might have been, at least in my opinion, better conducted at some places.

A sectione entitled “ The conceptual framework” is added to revised manuscript in Page 13.

3. Please check again if all figures/tables are referenced in the paper. 

All the tables and figures are refrenced (where applicable) in the manuscript.

4. Some figures (e.g., Figure 4) should be improved in the sense of informative content, meaning that the reader immediately understands the main point without even looking at the corresponding text.

Figure 4 is a snapshot of the simulation animation playback which shows the buses movement on a route. More information is added on the figure manualy to make it more informative. 

Reviewer 1 (major revision)

The paper deals with the topic of bus service reliability, improvement strategies and simulation model for evaluation of proposed strategies. Authors provided another view on topic, however I disagree with the statement that there is a gap in this field. Other authors and transportation companies just used different measurements to evaluate the service reliability. It would be usefull to compare proposed methodologies and strategies with proposals of other authors. The topic is still up-to-date and the proposed methodology and simulations are interesting, however I have few comments and recommendations for the authors.

We would like to express our appreciation for the time and effort devoted to the reviewing of our manuscript.

1- English language in the paper should be corrected. It contains mistakes and misprints.

We sent the manuscript to an English Editor. Authors believe that English sentences are improved in revised manuscript.

2- The style of references in the paper should be unified. Authors use various styles.

Thank you for this comment. The refrences are revised and corrected in a unified style.

3- Table 1 mentioned on the line 90 does not exist. Table 1 in the paper contains different information.

Based on our last revision on paper, we have decided to eliminate the “Table 1”. However, unfortunately we forgot to remove this sentence on line 90. This mistake is coorected in revised manuscript (the sentence is deleted). 

4- Are the values in the table 2 correct? The length of the line in the westbound direction is twice as long as in the eastbound direction?

Thank you for this comment. The route length is corrected in revised manuscript.

5- The model from the description in the paper seems to be very simple. It would be usefull to describe the simulation model in more details (describe more the used data, describe the simulation tool used for implementing proposed model, etc.).

A section entitled “ The conceptual framework” is added to revised manuscript in Page 13.

6- The "cov" in the formula (2) should be described in more details.

As mentioned in line 534 (in revised manuscript), Cov(h) is the coefficient of variation of headways, which is very common statistical measure of the dispersion of data points in a data series around the mean. The coefficient of variation represents the ratio of the standard deviation to the mean, and it is a useful statistic for comparing the degree of variation from one data series to another, even if the means are drastically different from one another. CV can be calculated simply by dividing Standard Deviation over Mean of a sample. Most of the programing languages and statistical programs are able to calculate CV automatically. CV=σ/μ 

Authors believe that explanation on statistical parameters such CoV are not necessary in this transportation research paper. However, paragraph above, Cov equation and more details can be added to main body of revised manuscript, based on reviewer opinion.

7- Notation in formula (3) should be described in more details.

EWT = AWT ‒ SWT (3)

Equation 4 shows an extended version of equation 3, which is also a good description. Excess Waiting Time is equal to Actual minus Scheduled Waiting Time. The Actual Waiting Time (AWT) and the Scheduled Waiting Time (SWT) both are calculated using the equation 2 for average waiting time [𝑤]. The only difference is in the headways. For calculating AWT, actual headways which were recorded and extracted from AVL data should be used. While in SWT, scheduled headways (according to service providers’ time table) should be used. 

The above explianation is added to revised manuscript in page 24, lines 542 to 547.

8- Numerical experiments contain only one bus line. It would be usefull to add a discussion about the influence of other factors on the travelling (more bus lines in the system, interchanges between lines and its influence on waiting times and overall travelling time, etc.)

As a matter of fact, this study proposes a novel fundamental simulation model in order to analyzing and evaluating bus service reliability at high-frequency operation. Simulating of a bus network (more than one bus route) is beyond the scope of this study. However:

1- All the data, codes, methods and packages developed in R studio are provided as supplementary files. Therefore, simulating bus network can be an interesting future study, based on our methods and findings.

2- All the data used in this study are real data sets which were collected from bus company archive. Therefore, effect of various factors such as interchanges and other factors affecting travel times, are actually the nature of these data sets. As an example, a comprehensive explanation and analysis is added to revised manuscript entitled “Pervious Running Time Model”, in Pages 18 and 19.

Reviewer 2 (rejected):

This paper developed a simulation model of bus service to improve reliability of high-frequency bus service. Four different types of strategies (schedulebased departure from terminal, headway-based departure from terminal, previous holding strategy, prefol holding strategy) were selected according to Route U32 (Kuala Lumpur) specifications. It showed that waiting time has been reduced significantly and headway regularity has been improved. My main concerns are:

Thank you for your time and effort for reviewing our paper.

1- This paper developed a simulation model, but it does not specifically explain how the random processes are simulated, such as customer's arrival process etc, and how the traffic/other conditions influence the random process. Also, could you provide other parameters in the experiment such as customer size etc.?

Thank you for this comment. A section is added to revised manuscript entitled “Conceptual framework”, which starts from page 13. All the random process is clearly explained in under different categories. Passenger arrival rates, boardings and alightings are comprehensively explained and analyzed under section “Passenger Demand Model”. Moreover, External factors such as accidents, traffic jams, weather and etc. are described in revised manuscript, under “Running Time Model” section. (Pages 18 and 19, Lines 407 to 444). 

Passenger activities and average daily passenger demand are also described in “Passenger Demand Model” section (Pages 15 to 18).

2- The paper defined the expected waiting time in presence of headway variation and excess waiting time on page 18, but why defining it this way? What's the intuition behind this definition and why it is a good way to evaluate the service reliability?

There is a simple method to calculate the expected waiting time: In normal frequency services, waiting time can be expected to be half of the headway. Operators use this formula in field to determine the expected waiting time, since it is easy and reliable. However, using this formula means you simply neglect the variations. The equation 2 is proposed and used by many studies to determine the more exact value of waiting time, by taking the headway variations in to the account. 

Passengers value the waiting time for urban transit almost twice that of in vehicle travel time, which makes it a crucial component of total trip time (van Oort & van Nes, 2009) [1]. Nevertheless, waiting time alone is not a useful parameter to evaluating the bus service reliability. We need a parameter, which actually compares the actual value of waiting time and planned value of waiting time, to clearly understand how far we are from our planed services. EWT is therefore the difference between AWT and SWT and represents the additional wait time experienced by passengers due to irregular spacing of buses.

According to TCQSM (2010), excess waiting time is a suitable parameter for understanding the service reliability. Moreover, the concept that bus service reliability should be measured from a passenger’s point of view has been adopted by Transport for London (TfL) who use EWT as the key indicator of reliability for high frequency services (e.g. a route which has five or more buses per hour) [2][3].

3- Following concern (2), the term mean(h_actual) and cov(h_{actual}) are correlated, so how do you calculate EWT specifically in simulation?

Most of statistical terms are pre-defined in programing languages (such as Coefficient of Variation and Average values). However, we can easily define any additional terms in R studio. Waiting time should be calculated for each key stop separately in a uniform time of the day (like AM peak hours). Simulation model (like other software such as excel) follows the equation. Therefore, first average of recorded headways for a specific period of time (AM peak) and specific location (a key stop) will be calculated. Then using formula =σ/μ , simulation will calculates the CoV of headway. 

4- What's the running time for each simulation test?

“Running Time Model” section is added to revised manuscript (Pages 18 and 19, Lines 407 to 444). Basically, running times for each segment can be calculated from AVL data sets which are provided as support files for this study. For example, Running Time for segment 1 (distance between key stop 1 and 2 is considered as segment 1) is equal to Arrival time at Key stop 2 minus Arrival time at Key stop 1 minus Dwell time at key stop 1. Thousands of running times can be calculated using these AVL data sets. Running times were separated and modeled regarding to their location and time. The model is validated using wil-coxon test. After validation, this model is used to generate running time in simulation for each segment. The Running Time model developed by Moosavi and Yuen (2020) and Moosavi et al. 2017 was used for analyzing and generating running times [4,5].

5- Usually there would be multiple routes going through one stop, so could this framework/R package be extended to a traffic network?

As a matter of fact, this study proposes a novel fundamental simulation model in order to analyzing and evaluating bus service reliability at high-frequency operation. Simulating of a bus network (more than one bus route) is beyond the scope of this study. However, all the data, codes, methods and packages developed in R studio are provided as supplementary files. Therefore, simulating bus network can be an interesting future study, based on our methods and findings.

6- Is there a way to verify the small gap between the simulation result and the real-world implementations?

The simulation model is verified and validated as explained in pages 21 and 22, lines 468 to 496. The validation results approved that simulation model outputs are close enough to real world situation that we can consider them as a reliable results. Wilcoxon test results show the difference between models output and real world data. This test is part of simulation model and after each run the Wilcoxon test detects even small differences between real world and simulation results.

Reviewer 3 (major revision):

1. INTRODUCTION:

1.1. Lines 52 -73 does not represent an introduction into the subject matter rather it is a brief on what each section of their paper addresses. Author may choose to give a better introductory section on the subject matter then use between 3 to 5 lines to summarize the structuring of their paper.

Thank you for reviewing our paper and your valuable comments. Introduction is revised and rewritten according to this comment. We tried to present more introductory on objective and subject of the study. Revised Introduction can be found in Pages 3 and 4, Lines 55 to 79.

2. BODY AND RESULT:

2.1. Citation style on line 78 should consistent. Why the use of author names and reference number? 

Thank you for this comment. This error is corrected in revised version of manuscript.

2.2 On line 82, the year of publication for "TCQSM" should be included so as to show currency of thought.

The publication year of TCQSM is added in revised manuscript: Line 103 in revised manuscript. 

2.3 Sentence on line 87 and 88 does not communicate a complete thought.

The mentioned sentence is revised and corrected as follow: 

“Chapter 27 in HCM (2000) outlines the quality evaluation of the following four main transportation modes in terms of: a) service frequency, b) hours of service, c) passenger load, and d) service reliability.” Page 5, Lines: 108 to 110

2.3 Generally, the discuss on the subject matter isn't adequate enough to communicate necessary thought by the author.

A Conceptual Frame work is added to manuscript in revised version. Moreover, more details on simulation model and methodology are added to manuscript. 

2.4 Line 127 - 130: Author did not include the basis and/or for selection of the four reliability indicators in the study. Presentation of reviewed literature needs to be reviewed to convey sequential flow of thought Line 379 also does not give any basis for selection of the four indicators. Author simply referred to literature that isn't cited.

We decided to replace this sentence from line 127-130 to the end of literature review section, Page 12, Lines 289 to 292 . According to review of current available literature, there is no consistency in reliability definition and indicators. Companies have their own definition of bus service reliability and they mostly neglect about the passengers’ perspective of reliability. Accordingly, four different reliability indicators were selected in this study to cover both passengers’ and operators’ perceptions of reliability: waiting time and on-board crowding level from passengers’ perspective, and headway regularity index (on-time performance) and bus bunching/big gap percentage from operators’ perspective.

2.5 Also, lines 133 - 143 refer to previous section of the paper that aren't explicit.

Thank you for this comment. The problem is rectified in revised manuscript.

2.6 Tables on lines 454 - 460 seem questionable, probably errors while entering values, otherwise, there is need to justify and/or explain the negative percentage values.

The negative values are correct and indicate the reduction in parameter value, after implementing a strategy. For example Table 4 last column presents the changes in E.W.T after implementing terminal dispatching strategy. According to this table, E.W.T reduced (improved) 51.17 % after implementing scheduled-based dispatching strategy. 

Further explanation is added to revised manuscript on Page 26, Line 590 to 592.

According to all reviews, the paper should be likely rejected. However, the Academic Editor does not necessarily share the same opinion. The reason is that the paper, at least from the perspective of the AE, without any doubt, brings certain novelties and contributions. Thus, the AE recommends the authors the following to increase the likelihood of the possible further consideration of the paper: I suggest to strictly follow all the comments of the reviewers and the AE.

Academic Editor DD

References: 

1. Van Oort N, van Nes R. Regularity analysis for optimizing urban transit network design. Public Transp. 2009;1: 155–168. 

2. van Oort N, van Nes R. Control of public transportation operations to improve reliability: Theory and practice. Transp Res Rec J Transp Res Board. 2009; 70–76. 

3. Ap. Sorratini J, Liu R, Sinha S. Assessing bus transport reliability using micro-simulation. Transp Plan Technol. 2008;31: 303–324. 

4. Moosavi SMH, Ismail A, Balali A. Evaluating bus running time variability in high-frequency operation using automatic data collection systems. Pertanika J Sci Technol. 2017;25. 

5. Moosavi SMH, Choon Wah Y. Measuring Bus Running Time Variation during High-Frequency Operation Using Automatic Data Collection Systems. Inst Transp Eng ITE J. 2020;90: 45–49.

---

## [Editor Report · Decision Letter 1]

22 Apr 2020

Using Simulation Model as A Tool for Analysing Bus Service Reliability and Implementing Improvement Strategies

PONE-D-20-00591R1

Dear Author,

We are pleased to inform you that your manuscript has been judged scientifically suitable for publication and will be formally accepted for publication once it complies with all outstanding technical requirements.

With kind regards,

Dejan Dragan, PhD

Academic Editor

PLOS ONE

Additional Editor Comments (optional):

Editor's comments to the paper (round1):

Using Simulation Model as A Tool for Analysing Bus Service Reliability and Implementing Improvement Strategies

The primary objective of this research is to improve the reliability of high-frequency bus service and simulation tools currently used in the public transportation companies. Therefore, a simulation model of high-frequency bus service was developed in order to study the strategies to alleviate it. This study was designed to cover gaps that have been recognized in the literature. According to the authors best knowledge, currently, there is no bus service simulation package available in order to 1) Analyse and measure the level of bus service reliability considering both passengers, and agencies point of view, 2) Implementing corrective strategies and combinations of strategies on bus routes to find out the effect of different strategies, 3) Capturing and comparing the level of reliability before and after implementing any changes on the route 4) Illustrating the movement of buses on a specific route and number of onboard passengers in animation playback and 5) Analysing "Headway Regularity" in term of regularity index, big gap and bunching and excess waiting time. Since all simulation codes and Rstudio files are published with this paper, this simulation model can be expanded by other researchers for further studies in the future. The subject of this research is up-to-date and fundamentally interesting for scholars and practitioners from the field. Thus, from this point of view, the paper likely might have deserved an opportunity to be considered for a possible publication. More importantly, we are dealing with an open-source R environment, which is gradually becoming a leading software environment not only for the simulation but for all kinds of different analyses, modeling spheres, and many other tasks.

From the Editor’s point of view, the paper has been substantially improved while doing the corrections. All significant issues have been appropriately corrected, and comments have been adequately followed. Moreover, all the Reviewers’ questions and dilemmas have been satisfactorily explained. Maybe there is only one remaining issue that should be perhaps taken into consideration. It is maybe unusual that figures 10 and 11 appear in conclusion. Otherwise, the AE believes that the paper might have been considered to be accepted and proceeded in the further publishing process.

Academic Editor DD
---

## [Editor Report · Acceptance letter]

27 Apr 2020

PONE-D-20-00591R1 

Using Simulation Model as a Tool for Analyzing Bus Service Reliability and Implementing Improvement Strategies 

Dear Dr. Moosavi:

I am pleased to inform you that your manuscript has been deemed suitable for publication in PLOS ONE. Congratulations! Your manuscript is now with our production department. 

With kind regards,

on behalf of

Dr. Dejan Dragan 

Academic Editor

PLOS ONE